# Training-free Diffusion Model Alignment with Sampling Demons

**Po-Hung Yeh[1], Kuang-Huei Lee[2], Jun-Cheng Chen[1]**
[1]Academia Sinica, [2]Google DeepMind
`{pohungyeh, pullpull}@citi.sinica.edu.tw, leekh@google.com`

## Abstract

Aligning diffusion models with user preferences has been a key challenge. Existing methods for aligning diffusion models either require retraining or are limited to differentiable reward functions. To address these limitations, we propose a stochastic optimization approach, dubbed *Demon*, to guide the denoising process at inference time without backpropagation through reward functions or model retraining. Our approach works by controlling noise distribution in denoising steps to concentrate density on regions corresponding to high rewards through stochastic optimization. We provide comprehensive theoretical and empirical evidence to support and validate our approach, including experiments that use non-differentiable sources of rewards such as Visual-Language Model (VLM) APIs and human judgements. To the best of our knowledge, the proposed approach is the first inference-time, backpropagation-free preference alignment method for diffusion models. Our method can be easily integrated with existing diffusion models without further training. Our experiments show that the proposed approach significantly improves the average aesthetics scores for text-to-image generation. Implementation is available at `https://github.com/aiiu-lab/DemonSampling`.

## 1 Introduction

Diffusion models have been the state-of-the-art for image generation (Sohl-Dickstein et al., 2015; Ho et al., 2020; Song et al., 2021; Karras et al., 2022; Saharia et al., 2022; Rombach et al., 2022), but, commonly, the end users' preferences and intention diverge from the data distribution on which the model was trained. Aligning diffusion models with diverse user preferences is an ongoing and critical area of research.

One approach to aligning diffusion models with user preferences is to fine-tune using reinforcement learning (RL) to optimize the models based on rewards signals that reflect the user preferences (Black et al., 2023; Fan et al., 2023). However, retraining the model every time when the preference changes is computationally expensive and time-consuming.

An alternative approach is to guide the denoising process using a differentiable reward function. This can be done through classifier guidance at inference time (Dhariwal & Nichol, 2021; Wallace et al., 2023b; Bansal et al., 2024; Yoon et al., 2023) or backpropagation at training time (Prabhudesai et al., 2024; Clark et al., 2024; Xu et al., 2023). These methods are generally less resource-demanding and more efficient. While these methods are generally more efficient, they require the reward function to be differentiable. This limits the types of reward sources that can be used, as it excludes the non-differentiable sources like third-party Visual-Language Model (VLM) APIs and human judgements.

To address these limitations, we propose *Demon*, a novel stochastic optimization approach for preference optimization of diffusion models at inference time. Demon is a metaphor from Maxwell's Demon, an imaginary manipulator of natural thermodynamic processes. The core ideas are: (1) Quality of noises that seed different possible backward steps in a discretized reverse-time Stochastic Differential Equation (SDE) can be evaluated given a reward source; (2) Such evaluation enables us to synthesize "optimal" noises that theoretically and empirically improve the final reward of the generated image through stochastic optimization. Specifically, we leverage Probability Flow Ordinary Differential Equation (PF-ODE) (Song et al., 2021) or Consistency Model (CM) (Song et al., 2023;

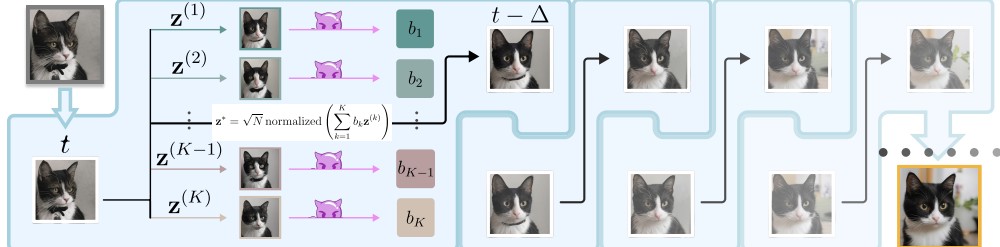

Figure 1: **Illustration of Demon.** Given a reverse-time SDE for denoising and an interval $[t_{\max}, t_{\min}]$, we first discretize it into $T$ steps, $t_{\max} > \cdots > t > t - \Delta > \cdots > t_{\min}$. At every reverse-time denoising step, from $t$ to $t - \Delta$, we synthesize an "optimal" noise $\mathbf{z}^*$ from $K$ i.i.d. noises w.r.t a given reward source and use $\mathbf{z}^*$ to seed the step. This enables guiding the denoising process towards generating images that are more aligned with the reward source and the preference that the reward source represents. More details are presented in Section 4.

Luo et al., 2023) to help us efficiently evaluate the possible backward steps, seeded with different Gaussian noises.

Our key contributions are summarized as follows:

- Our approach enables the use of reward signals in the denoising process regardless of whether the reward function is differentiable. This allows for the incorporation of previously inaccessible reward sources, such as VLM APIs. To the best of our knowledge, this is the first inference-time, backpropagation-free preference alignment method.

- Our method can be easily integrated with existing diffusion models in a plug-and-play fashion without retraining or fine-tuning.

- We provide a theoretical explanation for why our approach can improve the given reward function for image generation, which can be exploited for tuning hyperparameters.

- We demonstrate that our approach significantly improves the average aesthetics score (LAION, 2023) of Stable Diffusion models, achieving averages well above 8.0 compared to the Best-of-N random sampling upper bounds of 6.5 for SD v1.4 and 7 for SDXL. This improvement is achieved across various text-to-image generation tasks using prompts from prior work (Black et al., 2023), without relying on backpropagation-based preference alignment or model retraining.

## 2 RELATED WORK

**Diffusion Model.** Diffusion models for data generation were first proposed by Sohl-Dickstein et al. (2015), further developed for high-fidelity image generation by Ho et al. (2020), and generalized by Song et al. (2021) through the lens of SDEs. Karras et al. (2022) comprehensively studied the design space of Diffusion SDEs. In this work, we base many of the derivations on theirs. Furthermore, we focus on evaluating our method in the text-to-image generation setting (Rombach et al., 2022; Ho & Salimans, 2021; Podell et al., 2024)

**Human Preference Alignment.** Aligning models with human preferences has been studied with several approaches:reinforcement learning-based policy optimization (Fan et al., 2023; Yang et al., 2024; Black et al., 2023); training with reward backpropagation (Clark et al., 2024; Xu et al., 2023); backpropagation through the reward model and the diffusion chain (Prabhudesai et al., 2024; Wallace et al., 2023b; Bansal et al., 2024; Yoon et al., 2023). Many metrics and benchmarks for evaluating alignment has also been proposed, including those by Xu et al. (2023); Kirstain et al. (2023); LAION (2023); Wu et al. (2023), and we use these either as optimization objectives or evaluation of the

generated image. In Table 1, we further provide detailed comparisons of the proposed Demon approach with relevant existing methods in the literature from different aspects.

Table 1: A detailed comparison of different methods along various dimensions, including the ability to generalize to an open vocabulary, the necessity of a backpropagation signal for optimization, the method's capacity to avoid mode collapse and ensure distributional guarantees (Divergence Control). Our proposed method stands out for its zero-shot learning capabilities.

| Type | Methods | Open Vocab | Non-Backprop Objective | Divergence Control |
|------|---------|------------|------------------------|--------------------|
| Training | DPOK (Fan et al., 2023) | × | ✓ | ✓ |
| Training | DDPO (Black et al., 2023) | × | ✓ | × |
| Inference | DOODL (Wallace et al., 2023b) | ✓ | × | × |
| Training | DPO (Wallace et al., 2023a) | ✓ | ✓ | ✓ |
| Training | DRaFT (Clark et al., 2024) | ✓ | × | × |
| Inference | Demon | ✓ | ✓ | ✓ |

## 3 PRELIMINARY

**Score-Based Diffusion Model.** We base our derivation on EDM (Karras et al., 2022). With a sampling schedule $\sigma_t = t$, we can write the reverse-time SDE sampling towards the diffusion marginal distribution as follows.

$$\mathrm{d}\mathbf{x}_t = \underbrace{\left[-t\nabla_{\mathbf{x}_t} \log p\left(\mathbf{x}_t, t\right) - \beta t^2 \nabla_{\mathbf{x}_t} \log p\left(\mathbf{x}_t, t\right)\right]}_{\mathbf{f}_\beta(\mathbf{x}_t, t)} \mathrm{d}t + \underbrace{\sqrt{2\beta}t}_{g_\beta(t)} \mathrm{d}\omega_t, \tag{1}$$

where $p(\mathbf{x}_t, t) = p(\mathbf{x}_0, 0) \otimes \mathcal{N}\left(\mathbf{0}, t^2 \boldsymbol{I}_n\right)$ and $\otimes$ denotes the convolution operation. $\mathbf{x}_0$ is a clean sample, $\mathbf{x}_0 \sim p_{\text{data}}$, and $\mathbf{x}_t$ is a noisy sample at time $t$. $\beta$ expresses the relative rate at which existing noise is injected with new noise. In EDM, $\beta$ is a function of $t$, but in our study, we set $\beta$ to a constant for all $t$ for simplicity. Essentially, $\mathbf{f}_\beta(\mathbf{x}, t)$ corresponds to drift and $g_\beta(t)$ corresponds to diffusion. As common in diffusion models, since $p(\mathbf{x}_t, t) \approx \mathcal{N}(\mathbf{0}, t^2 \mathbf{I}_N)$ for a large enough $t$, we sample $\mathbf{x}_{t_{\max}} \sim \mathcal{N}(\mathbf{0}, t_{\max}^2 \boldsymbol{I}_N)$ as the initial sample.

A comprehensive list of the notations and conventions used in this paper is provided at Appendix A.

## 4 REWARD-GUIDED DENOISING WITH DEMONS

In this section, we describe how Demon works in two steps: Section 4.1 explains the process of scoring Gaussian noises in reverse-time SDE with a reward function; Section 4.2 further explains how the noise scoring allows us to guide the denoising process to align with the reward function, which is what we refer to as *Demon*.

### 4.1 SCORING NOISES IN REVERSE-TIME SDE

Let $\mathbf{x}_0$ be the clean image corresponds to a $\mathbf{x}_t$ at time step $t$, say:

$$\mathbf{x}_0 = \mathbf{x}_t + \int_t^0 \mathbf{f}_\beta(\mathbf{x}_u, u) \,\mathrm{d}u + g_\beta(u) \,\mathrm{d}\omega_u\,, \tag{2}$$

where Equation (2) is denoted as $\mathbf{x}_0 \mid_\beta \mathbf{x}_t$, shorthanded as $\mathbf{x}_0 \mid \mathbf{x}_t$. For an arbitrary reward function $r$ e.g. aesthetics score, we define the reward estimate of $\mathbf{x}_t$ at time step $t$ as

$$r_\beta(\mathbf{x}_t, t) \coloneqq \mathbb{E}_{\mathbf{x}_0 \mid \mathbf{x}_t} \left[r(\mathbf{x}_0)\right]. \tag{3}$$

This can be estimated with a Monte Carlo estimator by averaging over the reward of several SDE samples, but it requires many sample evaluations for high accuracy. To address this weakness, we introduce an alternative estimator for $r_\beta(\mathbf{x}_t, t)$ based on PF-ODE (Song et al., 2021).

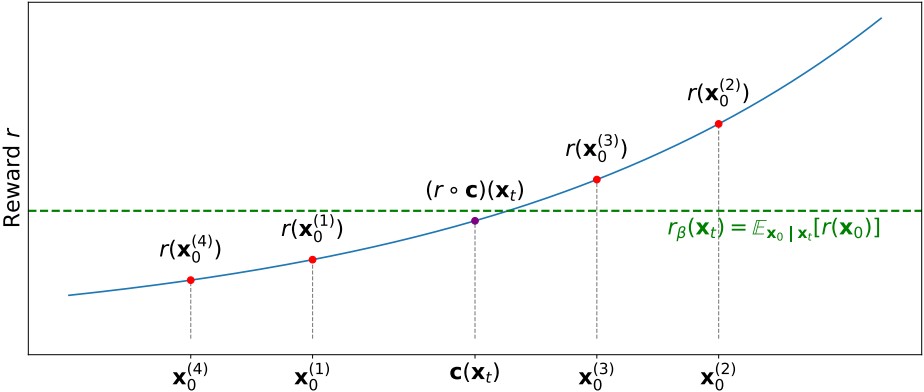

Figure 2: The illustration of the proximity between the $r_\beta$ and $r \circ c$. In this figure, the $\beta$ is nonzero and $r$ is near harmonic (i.e., $\nabla^2 r \approx 0$.). The red points indicate i.i.d. SDE samples and the purple ODE approximation of $\mathbf{x}_t$. The green line indicates the expectation of the rewards of the SDE samples (e.g., an approximate estimation, $\frac{1}{4}\sum_{i=1}^{4} r(\mathbf{x}_0^{(i)})$).

As shown in Song et al. (2021); Karras et al. (2022), the reversed-time SDE reduces to PF-ODE when $\beta \equiv 0$. For each $t$, a diffeomorphic relationship exists between a noisy sample $\mathbf{x}_t$ and a clean sample $\mathbf{x}_0$ generated by PF-ODE.

Similar to consistency models, with $\mathbf{x}'_t$ denoting an ODE trajectory instead of $\mathbf{x}_t$, we can denote this deterministic mapping from the domain of $\mathbf{x}_t$ to the domain of $\mathbf{x}_0$ as $\mathbf{c}(\mathbf{x}_t, t)$ as

$$\mathbf{c}(\mathbf{x}'_t, t) := \mathbf{x}'_0 = \mathbf{x}'_t + \int_t^0 \mathrm{d}\mathbf{x}'_u, \quad \text{where} \quad \mathrm{d}\mathbf{x}'_u = -u\nabla_{\mathbf{x}'_u} \log p\left(\mathbf{x}'_u, u\right)\,\mathrm{d}u. \tag{4}$$

Then, we can write $(r \circ \mathbf{c})(\mathbf{x}_t, t) = r(\mathbf{c}(\mathbf{x}_t, t))$ as the reward of the generated clean sample. This approximates $r_\beta(\mathbf{x}_t, t)$ using only one evaluated sample. In fact, we can characterize the difference between the approximate reward using ODE $(r \circ \mathbf{c})(\mathbf{x}_t, t)$ and the exact reward estimate using SDE $r_\beta(\mathbf{x}_t, t)$ as in Lemma 1. The right hand side of Equation (5) shows that, as $\beta \to 0$, the approximation becomes exact: $\lim_{\beta \to 0^+} r_\beta(\mathbf{x}_t, t) = (r \circ \mathbf{c})(\mathbf{x}_t, t)$. Intuitively, this result aligns with SDEs reducing to ODEs when $\beta$ approaches zero in image domains (Song et al., 2021).

**Lemma 1** (Itô Integral Representation of Reward Proximity Error. Proof is in Appendix D.1). *Let the reward estimate function, $h(\mathbf{x}_t, t) = (r \circ \mathbf{c})(\mathbf{x}_t, t)$, be shorthanded as $h$. We have:*

$$r_\beta(\mathbf{x}_t, t) - (r \circ \mathbf{c})(\mathbf{x}_t, t) = \mathbb{E}_{\mathbf{x}_0|\mathbf{x}_t}\left[\int_t^0 \nabla_{\mathbf{x}_u} h \cdot \mathrm{d}\mathbf{J}_\beta(\mathbf{x}_u, u) - \beta u^2 \nabla^2 h \,\mathrm{d}u\right]. \tag{5}$$

*where $\mathbf{x}_0$ is sampled from Equation (2), $\nabla^2 h$ is the Laplacian of $h$ and*

$$\mathrm{d}\mathbf{J}_\beta(\mathbf{x}_u, u) = -\beta u^2 \nabla_{\mathbf{x}_u} \log p\left(\mathbf{x}_u, u\right)\,\mathrm{d}u + \sqrt{2\beta}u\,\mathrm{d}\omega_u, \tag{6}$$

As demonstrated in Appendix D.1, Lemma 1 implies that when the Laplacian of the reward function is approximately zero ($\nabla^2 r \approx 0$), $r_\beta \approx r \circ \mathbf{c}$. We also illustrated the idea in Figure 2. For better presentation, we conveniently abbreviate $r_\beta(\mathbf{x}_t, t)$ as $r_\beta(\mathbf{x}_t)$, $\mathbf{c}(\mathbf{x}_t, t)$ as $\mathbf{c}(\mathbf{x}_t)$ and $(r \circ \mathbf{c})(\mathbf{x}_t, t)$ as $(r \circ \mathbf{c})(\mathbf{x}_t)$ in this paper.

### 4.2 DEMONS FOR REWARD-GUIDED DENOISING

In the section, we first outline the general pipeline of the proposed algorithm. Then, we introduce two approaches, *Tanh Demon* and *Boltzmann Demon*, to synthesize optimal noises for guiding reverse-time SDE solution; we show that the proposed methods optimize the final reward value with theoretical guarantee, essentially achieving alignment.

---

**Algorithm 1** A Numerical Step with Demon

1: **Input:** $\mathbf{x}_t, t, \Delta, K$
2: **Output:** $\hat{\mathbf{x}}_{t-\Delta}$
3: **for** $k = 1$ **to** $K$ **do**
4:     Draw $\mathbf{z}^{(k)} \sim \mathcal{N}(\mathbf{0}, \boldsymbol{I}_n)$
5:     $\hat{\mathbf{x}}_{t-\Delta}^{(k)} \leftarrow \text{heun}(\hat{\mathbf{x}}_t, \mathbf{z}^{(k)}, t, \Delta)$
6:     $R_k \leftarrow (r \circ \mathbf{c})(\hat{\mathbf{x}}_{t-\Delta}^{(k)})$ {implementing $r_\beta(\hat{\mathbf{x}}_{t-\Delta}^{(k)})$}
7: **end for**
8: $[b_k] \leftarrow \textbf{Demon}([R_k])$
9: $\mathbf{z}^* \leftarrow \sqrt{N} \text{ normalized}\left(\sum_{k=1}^{K} b_k \mathbf{z}^{(k)}\right)$
10: $\hat{\mathbf{x}}_{t-\Delta} \leftarrow \text{heun}(\hat{\mathbf{x}}_t, \mathbf{z}^*, t, \Delta)$
11: **Return** $\hat{\mathbf{x}}_{t-\Delta}$

---

Following Karras et al. (2022), an SDE numerical evaluation of $\hat{\mathbf{x}}_{t-\Delta}$ sampled from $\mathbf{x}_t$ can be seeded by noise $\mathbf{z}$ via a step of Heun's $2^{nd}$ order method (Ascher & Petzold, 1998) as follows:

$$\mathbf{z} \sim \mathcal{N}(\mathbf{0}, \boldsymbol{I}_n) \tag{7}$$

$$\hat{\mathbf{x}}_{t-\Delta} = \text{heun}(\mathbf{x}_t, \mathbf{z}, t, \Delta) \tag{8}$$

$$\coloneqq \mathbf{x}_t - \frac{1}{2}\left[\mathbf{f}_\beta(\mathbf{x}_t, t) + \mathbf{f}_\beta(\tilde{\mathbf{x}}_{t-\Delta}, t - \Delta)\right]\Delta + \frac{1}{2}\left[g_\beta(t) + g_\beta(t-\Delta)\right]\mathbf{z}\sqrt{\Delta}, \tag{9}$$

where $\mathbf{z}$ is a Gaussian noise, and $\text{heun}$ is the stochastic backward step from $\mathbf{x}_t$ to $\hat{\mathbf{x}}_{t-\Delta}$. The intermediate approximation $\tilde{\mathbf{x}}_{t-\Delta}$ is given by $\tilde{\mathbf{x}}_{t-\Delta} \coloneqq \mathbf{x}_t - \mathbf{f}_\beta(\mathbf{x}_t, t)\Delta + g_\beta(t)\mathbf{z}\sqrt{\Delta}$. While we use Heun's method here, other solvers can work too.

For image generation, Gaussian noise $\mathbf{z}$ is usually high-dimensional. For a high-dimensional $\mathbf{z}$, we can assume that it's likely on a $\sqrt{N}$ sphere (Lemma 5, Appendix). This allows us to weighted-combine various noises into a new noise $\mathbf{z}^*$:

$$\mathbf{z}^* = \sqrt{N} \text{ normalized}\left(\sum_{k=1}^{K} b_k \mathbf{z}^{(k)}\right), \tag{10}$$

where $\mathbf{z}^{(k)}$ are i.i.d. unit Gaussian noises, and $b_k$ are the search space. We outline the pseudocode of a numerical step with our proposed method in Algorithm 1. In the following, we describe the details of the proposed *Tanh Demon* and the *Boltzmann Demon* to determine the weights $b_k$.

**Tanh Demon.**    Intuitively, we may consider **up-weighting** the good noises that improve the reward and **down-weighting** the bad noises that harm the reward, compared to the average reward $\hat{\mu}$. As shown in Figure 3, Tanh Demon assigns positive weights to the good noises and negative weights to the bad noises with the $\tanh$ function, based on the reward estimates of the noises (Equation (5)) relative to the average $\hat{\mu}$ of $(r \circ \mathbf{c})(\hat{\mathbf{x}}_{t-\Delta}^{(k)})$:

$$\mathbf{z}^* = \sqrt{N} \text{ normalized}\left(\sum_{k=1}^{K} b_k^{\text{tanh}} \mathbf{z}^{(k)}\right), \quad \text{where} \quad b_k^{\text{tanh}} \leftarrow \tanh\left(\frac{(r \circ \mathbf{c})(\hat{\mathbf{x}}_{t-\Delta}^{(k)}) - \hat{\mu}}{\tau}\right), \tag{11}$$

where $\tau$ is the temperature parameter to $\tanh$, which can be adaptively tuned (as shown in Table 8). The average $\hat{\mu}$ is computed $\frac{1}{K}\sum_{k=1}^{K}(r \circ \mathbf{c})(\hat{\mathbf{x}}_{t-\Delta}^{(k)})$.

Under the assumption of our reward estimate proximity $r_\beta \equiv r \circ \mathbf{c}$, the Tanh Demon method is guaranteed to improve the final results, formalized in the following lemma:

**Lemma 2** (Improvement Guarantee of Tanh Demon. Proof in Appendix D.3). *Assume the truncation error terms in Equation* (36) *are negligible and* $r_\beta \equiv r \circ \mathbf{c}$. *Let* $\mathbf{z}^*$ *be derived from Equation* (11). *With probability* 1, $r(\hat{\mathbf{x}}_0^{\text{tanh}}) > r_\beta(\mathbf{x}_t)$, *where* $\hat{\mathbf{x}}_0^{\text{tanh}}$ *is derived by applying* $\mathbf{z}^*$ *on every step.*

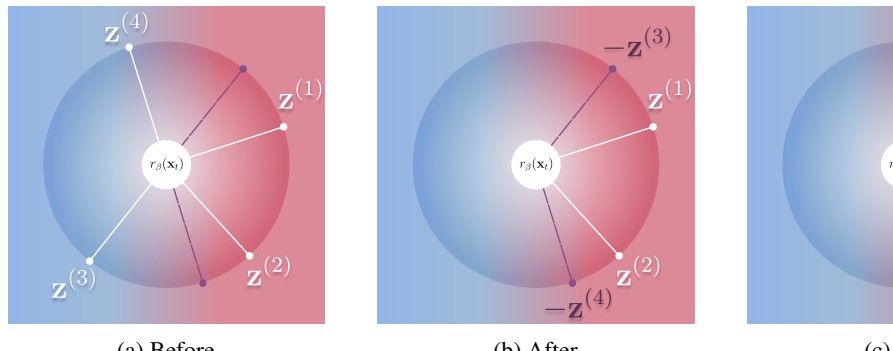

(a) Before        (b) After        (c) Final

Figure 3: An illustration of the Tanh Demon sampling method where $K = 4$. (a) A SDE step generates several samples, each determined by sampled noise $\mathbf{z}_k$. We use Tanh Demon to classify each noise sample as "low-reward" or "high-reward" w.r.t $r_\beta(\boldsymbol{x}_t)$ based on their respective reward estimates. (b) We penalize low-reward noise with $\tanh$ to multiply a negative weight which is equivalent to flipping the noise, (c) It shows how the post-processed noises are averaged and projected onto the high-dimensional sphere, resulting in a feasible noise representation $\mathbf{z}^*$ with high-reward estimate.

**Boltzmann Demon.** Another intuitive approach, equivalent to the single-step cross entropy approach (De Boer et al., 2005), is to estimate the candidate with **maximum reward**. We propose the Boltzmann Demon, which assigns noise weights as follows:

$$b_k^{\text{boltz}} \leftarrow \frac{\exp\left(r \circ \mathbf{c}(\hat{\mathbf{x}}_{t-\Delta}^{(k)})/\tau\right)}{\sum_{k=1}^{K} \exp\left(r \circ \mathbf{c}(\hat{\mathbf{x}}_{t-\Delta}^{(k)})/\tau\right)}, \tag{12}$$

where the Boltzmann distribution (i.e., $\mathrm{softmax}$ function) approximates **the behavior of the maximum function** as the temperature $\tau$ approaches to zero. The theoretical guarantee of improvement in $r_\beta$ in expectation is provided in Lemma 3, assuming $r_\beta \equiv r \circ \mathbf{c}$. Although empirically, we find that Tanh Demon outperforms Boltzmann Demon, adjusting $\tau$ in Boltzmann Demon provides control over deviation from the original SDE distribution, as demonstrated in Lemma 4 (Appendix).

### 4.3 COMPUTATIONAL CONSIDERATIONS

Let's first consider a Demon sampling trajectory $\mathbf{x}_{t_1} > \mathbf{x}_{t_2} > \cdots > \mathbf{x}_{t_T} \approx 0$ for a fixed number $T$. Each Demon's trajectory requires $\mathcal{O}(K \cdot T)$ evaluations of $\mathbf{c}$, and each evaluation comes with one reward estimation. The compute time is mainly influenced by the implementation of $r \circ \mathbf{c}$. We discuss two aspects of $r \circ \mathbf{c}$—the temporal cost and the fidelity—which are vital to the algorithm's time complexity and reward performance, respectively.

Note that Tanh or Boltzmann Demon itself does not strictly specify the implementation of $r \circ \mathbf{c}$; our default option uses Heun's ODE solver, but using a Consistency Model (CM) distilled from the original diffusion model significantly accelerates computation. An alternative, which we refer as Tanh-C, is to combine our Tanh Demon algorithm with an off-the-shelf CM to implement $r \circ \mathbf{c}$. While using Tanh-C may slightly degrade the results due to the fidelity loss from using a CM (see Table 2), this approach is particularly effective when faster results are required since the computation of $\mathbf{c}$ is much quicker. For a larger $T$, however, the default Tanh Demon using Heun's method outperforms Tanh-C in terms of reward performance.

As shown in Table 10, using the text-to-image generation task settings from Black et al. (2023), the Demon algorithm achieves an aesthetics score of $6.72 \pm 0.26$ on SD v1.4, requiring 5 minutes (i.e., $K = 16, T = 16$) on an NVIDIA RTX 3090 GPU. Within the same 5-minute computation window, the Tanh-C variant achieves an improved score of $7.27 \pm 0.33$ (i.e., $K = 16, T = 64$). Notably, the upper bound for randomly sampled SD v1.4 is approximately 6.5, obtained after more than 10 minutes and 800 reward function queries. See Appendix B for parameter guidelines and settings.

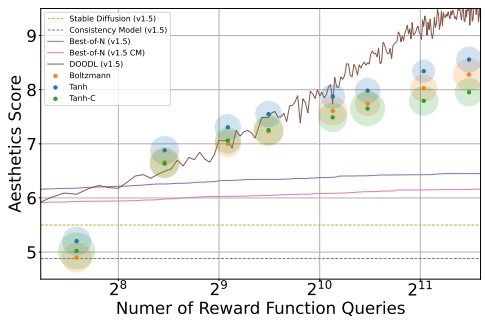 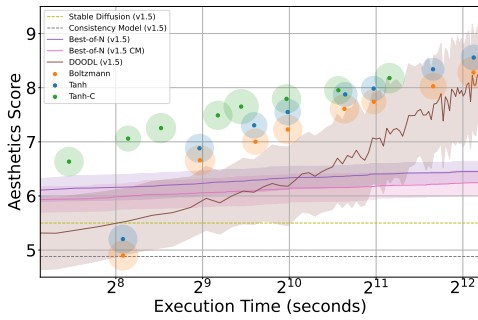

(a) Performance w.r.t Reward Query Number

(b) Performance w.r.t Execution Time

Figure 4: Performance comparison of the proposed algorithm and other baseline methods in terms of the number of reward queries and execution time; the dependent variable is $T$, which is suggested to be larger for SDE solver to reduce truncation error. Although DOODL can achieve similar results to ours, it relies on reward backpropagation, whereas our backpropagation-free methods do not require this. The shaded areas and the radii of solid circles represent the standard deviation of the evaluation results.

## 5 EXPERIMENTS

In this section, we present both quantitative and qualitative evaluations of our methods. Due to the page limit, we include the details of the implementation and experimental settings in Appendix I and the subjective results in Appendix G.2.

**Baseline Comparison.** For the performance comparisons between our method and other baselines, we use the LAION (2023) aesthetics scores (Aes) as the evaluation metric, and the scores are evaluated on a set of various prompts for generating animal images, which were from the full set of 45 common animals in ImageNet-1K (Deng et al., 2009), created by Black et al. (2023). We use 20-step Heun's ODE for reward estimate for our methods and Best-of-N (SD v1.5).

In terms of reward queries, Tanh/Tanh-C outperforms other baseline methods in most cases, including our Boltzmann method and Best-of-N. Our methods are even comparable to the backpropagation-based DOODL (Wallace et al., 2023b), the state-of-the-art method optimized over the reward function. In terms of execution time, Tanh/Tanh-C consistently outperforms DOODL due to the exclusion of backpropagation; Tanh-C further benefits from its effective computational cost, given limited time. Moreover, our method's backpropagation-free nature makes it more resistant to reward hacking (Table 3). For further comparison on PickScore (Kirstain et al., 2023), please refer to Appendix E.1.

**Comparison of Reward Estimation Approaches.** Figure 4 demonstrates a comparison of the proposed methods with different $r \circ \mathbf{c}$ implementations, including 20-step Heun's ODE (Tanh) and 1-step CM (Tanh-C). **Tanh**, which uses a 20-step ODE for accurate ($r \circ \mathbf{c}$), consistently outperforms the Best-of-N baseline given an equivalent number of reward queries. **Tanh-C**, which employs a 1-step CM for fast reward evaluation, outperforms Tanh when considering limited execution time. These observations suggest that the quality of $r \circ \mathbf{c}$ indeed plays a significant role in the effectiveness of our method.

To further validate the importance of $r \circ \mathbf{c}$, we conduct a comparative analysis based on Lemma 1 ($r \circ \mathbf{c} \approx r_\beta$). In this analysis, we evaluate the accuracy and computational cost across three methods: 20-step Heun's ODE, 4-step Heun's ODE, and 1-step CM; both diffusion and consistency models are based on the SD v1.5 and distilled by Luo et al. (2023). Experiments were performed with $t = 0.5, 3.0, 1.0, 7.0, 14.0$ ranging from 0.002 to 14.648. Accuracy was quantified using the standard of $r_\beta(\mathbf{x}_t) - (r \circ \mathbf{c})(\mathbf{x}_t)$. Here, $\mathbf{x}_t$ is sampled from $\mathcal{N}(\mathbf{0}, t_{\max}^2 \mathbf{I}_n)$ and integrated from $t_{\max}$ to $t$ using a 200-step diffusion model ODE, performed on the full set of Black et al. (2023).

The results, presented in Table 2, support that the quality of $r \circ \mathbf{c}$ influences both the algorithm's speed and reward performance. For the ODE methods, the trend follows our expectation: As $t$ approaches 0, the standard deviation decreases, which can be attributed to the diminishing noise

Table 2: Comparison of accuracy and time cost across different $r \circ \mathbf{c}$ implementations using the full set of animal prompts Black et al. (2023).

| Implementation | Time (s) | Standard Deviation of $r_\beta - r \circ \mathbf{c}$ | | | | |
|---|---|---|---|---|---|---|
| | | $t = 0.5$ | $t = 1.0$ | $t = 3.0$ | $t = 7.0$ | $t = 14.0$ |
| 20-step ODE | 1.94 | $\mathbf{8.1 \times 10^{-2}}$ | $\mathbf{1.53 \times 10^{-1}}$ | $\mathbf{3.02 \times 10^{-1}}$ | $\mathbf{3.25 \times 10^{-1}}$ | $\mathbf{3.97 \times 10^{-1}}$ |
| 4-step ODE | 0.41 | $8.3 \times 10^{-2}$ | $1.82 \times 10^{-1}$ | $3.39 \times 10^{-1}$ | $3.52 \times 10^{-1}$ | $4.15 \times 10^{-1}$ |
| 1-step Consistency | $\mathbf{0.18}$ | $1.71 \times 10^{-1}$ | $2.64 \times 10^{-1}$ | $4.03 \times 10^{-1}$ | $3.85 \times 10^{-1}$ | $4.85 \times 10^{-1}$ |

as the posterior $p(\mathbf{x}_t \mid \mathbf{x}_0)$ becomes more sharply peaked; the number of ODE steps is crucial to the quality of the generated outputs; more steps generally lead to higher fidelity results, although this comes at the cost of increased computational time; using 1-step CM leads to inferior results compared to using ODE, supposedly as the distillation gap and the limited model capacity result in lower-fidelity reconstructions.

Table 3: Results using various reward functions and different generation methods. Each column represents a specific reward objective, with the best performance highlighted in bold.

| Generation method | Aes ↑ | IR ↑ | Pick ↑ | HPSv2 ↑ | Time |
|---|---|---|---|---|---|
| SD v1.4 | $5.34 \pm 0.56$ | $-0.00 \pm 0.95$ | $0.202 \pm 0.008$ | $0.216 \pm 0.036$ | 5 s |
| DPO | $5.36 \pm 0.72$ | $0.03 \pm 0.84$ | $0.203 \pm 0.007$ | $0.229 \pm 0.027$ | 5 s |
| Uni (CLIP-guided) | $4.11 \pm 0.74$ | $-1.81 \pm 0.50$ | $0.191 \pm 0.014$ | $0.173 \pm 0.022$ | 55 min |
| Tanh + Aes | $\mathbf{7.35 \pm 0.40}$ | $-0.03 \pm 1.24$ | $0.211 \pm 0.010$ | $0.257 \pm 0.041$ | |
| Tanh + IR | $5.96 \pm 0.28$ | $\mathbf{1.95 \pm 0.07}$ | $0.216 \pm 0.012$ | $0.286 \pm 0.033$ | |
| Tanh + Pick | $6.14 \pm 0.48$ | $1.39 \pm 0.57$ | $\mathbf{0.245 \pm 0.010}$ | $0.312 \pm 0.033$ | 18 min |
| Tanh + HPSv2 | $5.98 \pm 0.45$ | $1.51 \pm 0.63$ | $0.228 \pm 0.011$ | $\mathbf{0.367 \pm 0.027}$ | |
| Tanh + Ensemble | $6.53 \pm 0.50$ | $1.81 \pm 0.15$ | $0.236 \pm 0.014$ | $0.356 \pm 0.030$ | |
| Best-of-N | $\mathbf{6.32 \pm 0.34}$ | $\mathbf{1.69 \pm 0.18}$ | $\mathbf{0.218 \pm 0.009}$ | $\mathbf{0.291 \pm 0.015}$ | 18 min |
| DOODL + Aes | $5.59 \pm 0.29$ | $-0.68 \pm 1.06$ | $0.197 \pm 0.008$ | $0.221 \pm 0.028$ | 18 min |
| DOODL + Pick | $5.21 \pm 0.46$ | $-0.12 \pm 0.84$ | $0.204 \pm 0.010$ | $0.220 \pm 0.035$ | 1.1 hr |

Table 4: Using Tanh Demons with various reward functions. The baseline, Stable Diffusion v1.4, refers to the standard model without our proposed enhancements.

| **Baseline** | **Best-of-N** | **Uni** | **DOODL** | **Aes** | **Ensemble** | **DPO** |
|---|---|---|---|---|---|---|

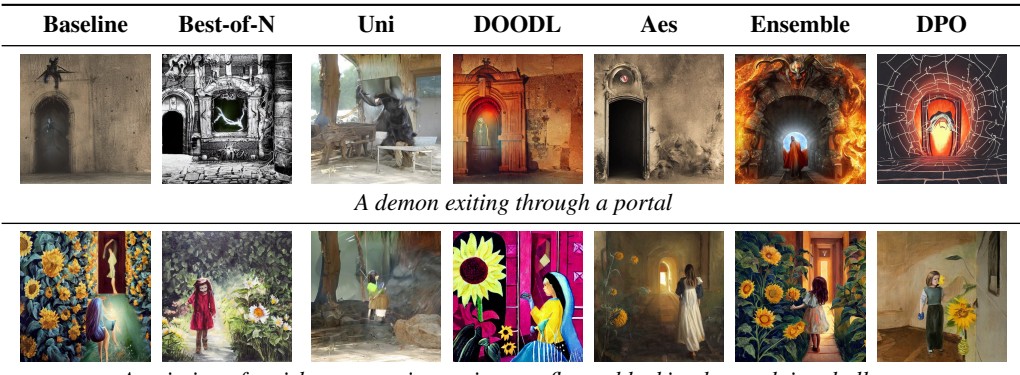

*A demon exiting through a portal*

*A painting of a girl encountering a giant sunflower blocking her path in a hallway*

**Image Generation with Various Reward Functions.** While our method optimizes a given reward function, as shown in Figure 4, we also present qualitative results in Table 4 and cross-validation results in Table 3. These results demonstrate perceptual preferences by averaging rewards derived from prompts provided in Tables 12 and 15.

We employ our Tanh Demon with various reward functions, such as Aes (LAION, 2023), ImageReward (IR)(Xu et al., 2023), PickScore (Pick)(Kirstain et al., 2023), HPSv2 (Wu et al., 2023), and a scaled sum (Ensemble) of Aes, IR, Pick, and HPSv2. For comparison, we include the best reward during sampling under the same time condition (best-of-N) and DOODL (Wallace et al., 2023b), which is optimized on Aes and Pick to modify results generated by PF-ODE using their recommended settings. For reference, we also provide the performance of a training-based method, DPO Wallace

et al. (2023a), and a backpropagation-based method, Universal Guidance (Uni) by Bansal et al. (2024), guided by the CLIP condition.

Starting from the baseline SDv1.4 under similar computational conditions, Tanh exhibits improvement **across the four metrics**, demonstrating robustness even when acknowledging slight over-optimization of the objective—in contrast to DOODL. By comparing best-of-N and Tanh with the Ensemble reward, our method achieves superior performance on *each* objective using the Ensemble, demonstrating not only the ability to integrate a mixture of rewards but also generating a **superior samples** that outperforms all individual best samples selected by best-of-N methods.

**Alignment with preferences of VLMs (Non-differentiable).** In Table 5, we present qualitative results of aligning diffusion models SDXL to preferences of VLMs from API, as a demonstration of using non-differentiable reward sources. In this experiment, we use Google Gemini Pro v1.0 (Gemini Team Google, 2024) and GPT4 Turbo (OpenAI, 2024). In each step, the VLM receives a fixed prompt, e.g. "You are a journalist who wants to add a visual teaser for your article to grab attention on social media or your news website", and is asked to select the best-matching intermediate sample from generated images. VLMs are presented with $\mathbf{c}(\mathbf{x}_t)$ and $\mathbf{c}(\hat{\mathbf{x}}_{t-\Delta}^{(k)})$ produced by PF-ODE. The reward $b_k^{\mathrm{VLM}}$ is $0.5$ if the VLM selects $\mathbf{c}(\hat{\mathbf{x}}_{t-\Delta}^{(k)})$ and $-0.5$ otherwise. We also use PickScore (Kirstain et al., 2023) to evaluate the results and find that 14 out of 16 images generated with VLMs show improvements compared to directly generating with PF-ODE.

Table 5: Using VLMs to generate images. PF-ODE (baseline) refers to a baseline without using our method for alignment. Columns 3-6 indicate the role that the agent plays in the given prompt.

| Model | Baseline | Teacher | Artist | Researcher | Journalist |
|---|---|---|---|---|---|
| Gemini-SD v1.4 | | | | | |
| Gemini-SDXL | | | | | |
| GPT-SD v1.4 | | | | | |
| GPT-SDXL | | | | | |

**Manual Selection.** We also explore using online interactive human judgements to guide diffusion. That means, the users themselves would be (non-differentiable) reward functions. We let users directly interact with our method to generate desired images. Figure 5a shows an example interface created by us for an image resembling a given reference cat image. At each iterative step from $t$ to $t - \Delta$, we sample 16 i.i.d. copies of $\boldsymbol{x}_{t-\Delta}$ and compute $\mathbf{c}(\boldsymbol{x}_{t-\Delta})$ with PF-ODE. The user then manually select their preferred image, assigning a reward of $+1$ to it and $-1$ to the others. We continue this process until there is no obvious preferred ones among the generated images. As shown in Figure 5b, the image generated by our method more closely matches the target than the one produced by PF-ODE. We also measure the improvement with DINOv2 (Oquab et al., 2023) embedding cosine similarity

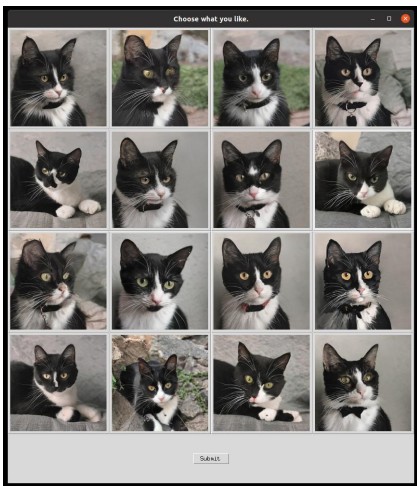 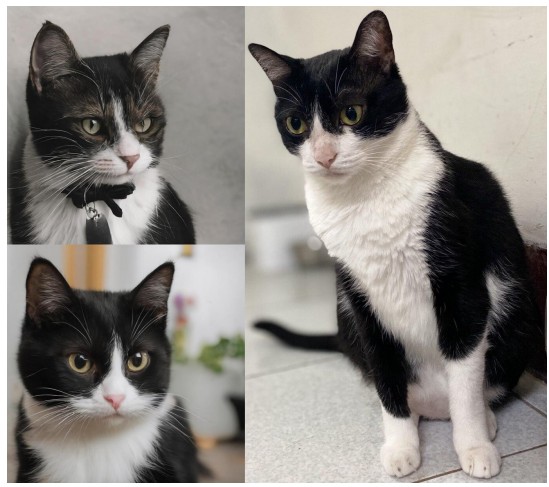

(a) Our user interface for interacting with our algorithm (0.594 cosine similarity).

(b) (Top Left) Image generated by PF-ODE (0.622 cosine similarity). (Bottom Left) Image generated by our method (0.708 cosine similarity). (Right) Reference image.

Figure 5: We design an application for manual interaction with our algorithm. Our author selects the images, and the criteria are based on the author's preference (non-preferred images are kept unselected), where the author tries to align the reference image. We evaluate performance by measuring the cosine similarity of DINOv2 features between the targeted and reference images.

between the reference image and the generated image, and observe that the similarity improves from 0.594 to 0.708 through online user interactions.

## 6 CONCLUSION

This work addresses the challenge of better aligning pre-trained diffusion models without training or backpropagation. We first demonstrate how to estimate noisy samples' rewards based on clean samples using PF-ODE. Additionally, we introduce a novel inference-time sampling method, based on stochastic optimization, to guide the denoising process with any reward sources, including non-differentiable reward sources that includes VLMs and interactive human judgements. Theoretical analysis and extensive experimental results validate the effectiveness of our proposed method for improved image generation without requiring additional training. Through comprehensive empirical and theoretical analysis, we observe that the quality and efficiency of reward estimation $r \circ \mathbf{c}$ are essential for our algorithm, especially in balancing computational speed and reward performance.

## ACKNOWLEDGEMENT

This research is supported by National Science and Technology Council, Taiwan (R.O.C), under the grant number of NSTC-113-2634-F-002-007, NSTC-112-2222-E-001-001-MY2, NSTC-113-2634-F-001-002-MBK and Academia Sinica under the grant number of AS-CDA-110-M09. We specially thank Sirui Xie, I-Sheng Fang, and Jia-Wei Liao for the insightful discussions. We also extend our gratitude to Prof. Lam Wai-Kit, National Taiwan University, for his courses *MATH7509*, *MATH7510*, and *MATH5269*, which have significantly influenced this paper.

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

# A  NOTATIONS AND CONVENTIONS

Although we keep the main paper self-consistent, we provide this section to establish a consistent notation and convention for this paper as an aid.

## A.1  NOTATIONS

Table 6: Notations

| Notation | Description |
| --- | --- |
| $N$ | State dimension |
| $K$ | Noise sample number |
| $t_{\min}, t_{\max}$ | Upper bound and Lower bound of the noise level in numerical integration |
| $T$ | Number of time steps to solve SDE/ODE |
| $\beta$ | Noise parameter |
| $\mathbf{x}$ | State variable |
| $\mathbf{z}$ | Noise from Gaussian |
| $\Delta$ | Time step |
| $b_k$ | Unnormalized weight of noise |
| $\mathbf{f}_\beta$ | SDE policy drift |
| $g_\beta$ | SDE policy diffusion coefficient |
| $\mathbf{f}_0$ | PF-ODE policy drift |
| $\mathbf{J}_\beta$ | Langevin diffusion SDE |
| $\omega_t$ | reversed time Brownian motion |
| $r$ | Reward |
| $r_\beta$ | Reward estimates of SDE policy |
| $\mathbf{c}$ | Function to get expected ODE result |
| heun | Heuns's method, SDE solver for Karras SDE |

## A.2  CONVENTIONS

Table 7: Conventions

| Convention | Details |
| --- | --- |
| $r \circ \mathbf{c}$ | ODE reward estimate approximation, $r(\mathbf{c}(\mathbf{x}_t, t)) = (r \circ \mathbf{c})(\mathbf{x}_t, t)$ |
| $f \equiv g$ | For all $x$ of our interest, $f(x) = g(x)$ |
| $\hat{\mathbf{x}}$ | Numerical approximation with SDE solver |
| $\tilde{\mathbf{x}}$ | Intermediate value of Heun's method |
| $\mathbf{x}'$ | An ODE trajectory |
| $\tilde{\mathbf{z}}$ | Uniformly sampled from the sphere of radius $\sqrt{N}$ |
| $\mathbf{z}^*$ | Optimal noise generated by our algorithm |
| $\hat{\mu}$ | Mean of next state ODE reward estimates, $\frac{1}{K} \sum_{k=1}^K (r \circ \mathbf{c})(\hat{\mathbf{x}}_{t-\Delta}^{(k)})$ |
| $r(\mathbf{x}_t)$ | Shorthand for $r(\mathbf{x}_t, t)$ when the context is clear |
| $\mathbf{c}(\mathbf{x}_t)$ | Shorthand for $\mathbf{c}(\mathbf{x}_t, t)$ when the context is clear |
| $(r \circ \mathbf{c})(\mathbf{x}_t)$ | Shorthand for $(r \circ \mathbf{c})(\mathbf{x}_t, t)$ when the context is clear |
| $\mathbf{x}_0 \mid \mathbf{x}_t$ | Shorthand for $\mathbf{x}_0 \mid_\beta \mathbf{x}_t$, where $\mathbf{x}_0 = \mathbf{x}_t + \int_t^0 \mathbf{f}_\beta(\mathbf{x}_u, u)\, \mathrm{d}u + g_\beta(u)\, \mathrm{d}\omega_u$ |
| $\tilde{\omega}_t$ | Standard Brownian motion |

Instead of ODE, we sometimes use PF-ODE to highlight Song et al. (2021)'s contribution or when the context is unclear. They are equivalent in this paper.

## B  GUIDELINE ON PARAMETER SETTING

We explore the optimal setting for parameter $\tau$ with respect to the Boltzmann Demon and the Tanh Demon. For the Tanh Demon, the most effective $\tau$ is neither $\infty$ nor 0. We recommend setting $\tau$ to the standard deviation of the estimations $\{(r \circ \mathbf{c})(\boldsymbol{x}_{t-\Delta}^{(k)})\}_{k=1}^{K}$, rendering it an adaptive parameter that is robust to scaling. For the Boltzmann Demon, optimal performance is achieved by setting $\tau$ to 0, as demonstrated in Table 8.

Table 8: Comparison of performance for different settings of $\tau$ in the setting of Figure 4.

|  | $\tau = 1$ | $\tau = 0.01$ | Adaptive $\tau$ |
|---|---|---|---|
| Tanh | $7.40 \pm 0.30$ | $7.24 \pm 0.31$ | $\mathbf{7.45 \pm 0.33}$ |
| Boltzmann | $6.30 \pm 0.35$ | $\mathbf{7.28 \pm 0.30}$ | $6.85 \pm 0.37$ |

We also conduct an ablation study on the remaining parameters $K$ and $\beta$. The base configuration is $K = 16, \beta = 0.1$, with an adaptive temperature $\tau$ for the Tanh Demon. We set $T = 32$ for the ablation study of $\beta$ and $T = 64$ for $K$.

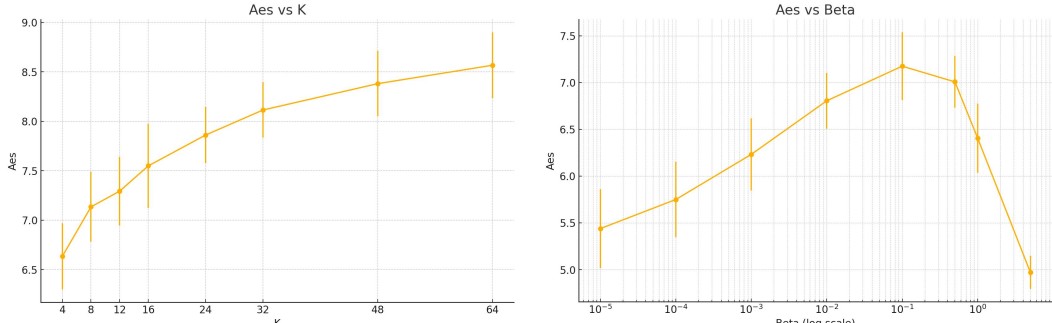

Figure 6: Comparison of our algorithm with respect to $K$ and $\beta$

We found a large $\beta$ makes the sampling unstable, given the number of steps $T$ is fixed. Predictably, sampling with a $\beta$ close to 0 is reduced to ODE. From our theoretical result Lemma 1, the design methodology, and empirical results, the guidelines Table 9 can assist users in setting parameters. We provide a sparse parameter search in Table 10.

| Parameter | Description |
|---|---|
| $K$ | Controls the noise distribution bias, positively affecting final quality and linearly increasing computational time. |
| $\beta$ | Adjusts the distribution's proximity to the original PF-ODE. Set empirically based on $r$'s characteristics. Lemma 1 suggests smaller $\beta$ for reward functions with Laplacian deviations. |
| $T$ | Inherit the properties of time steps $T$ from diffusion models, scaling computational time linearly. Karras's EDM recommends $T > 17$. |
| $\tau$ | Recommended values vary for Boltzmann and Tanh Demons, as detailed in Table 8. |
| $r \circ \mathbf{c}$ | Accurate reward estimates are critical for ensuring high final quality. |

Table 9: Guidelines for Setting Hyperparameters

Table 10: This table presents the experimental configurations used to measure aesthetics score under various animal prompts, presenting a sparse search of parameters. The time column represents the duration required to generate each image. We alias adaptive temperature as Adaptive.

| Demon | Checkpoint | $\beta$ | $K$ | $T$ | $\tau$ | Aes | Time (min) |
|---|---|---|---|---|---|---|---|
| Boltzmann | SD v1.4 | 0.1 | 16 | 64 | Adaptive | $6.408 \pm 0.36$ | 17.6 |
| | | | | | 1e-10 | $7.111 \pm 0.32$ | 16.6 |
| | SDXL | 0.05 | 16 | 32 | Adaptive | $6.853 \pm 0.37$ | 45.8 |
| | | | | | 1e-02 | $7.276 \pm 0.30$ | 45.4 |
| | | | | | 1 | $6.300 \pm 0.35$ | 46.1 |
| | | 0.1 | 16 | 64 | Adaptive | $6.990 \pm 0.38$ | 94.2 |
| | | | | | 1e-10 | $7.501 \pm 0.31$ | 93.1 |
| Tanh | SD v1.4 | 0.05 | 16 | 16 | Adaptive | $6.723 \pm 0.26$ | 5.0 |
| | | | | 32 | Adaptive | $7.073 \pm 0.22$ | 9.7 |
| | | | | 64 | Adaptive | $7.394 \pm 0.29$ | 18.7 |
| | | 0.1 | 16 | 64 | Adaptive | $7.549 \pm 0.43$ | 18.7 |
| | | | 64 | 64 | Adaptive | $8.566 \pm 0.33$ | 79.1 |
| | Diffusion-DPO | 0.1 | 16 | 64 | Adaptive | $7.564 \pm 0.34$ | 94.5 |
| | | 0.01 | 16 | 16 | Adaptive | $6.876 \pm 0.40$ | 22.0 |
| | SDXL | 0.05 | 16 | 16 | Adaptive | $6.866 \pm 0.35$ | 21.9 |
| | | | | 32 | Adaptive | $7.459 \pm 0.33$ | 46.0 |
| | | | | | 1e-02 | $7.244 \pm 0.31$ | 46.0 |
| | | | | | 1 | $7.398 \pm 0.30$ | 46.2 |
| | | 0.1 | 8 | 64 | Adaptive | $7.446 \pm 0.37$ | 47.0 |
| | | | 16 | 64 | Adaptive | $7.841 \pm 0.32$ | 94.4 |
| | | | 32 | 64 | Adaptive | $8.179 \pm 0.35$ | 188.8 |
| | | 0.5 | 16 | 32 | Adaptive | $6.370 \pm 0.35$ | 46.0 |
| Tanh-C | SD v1.4 | 0.5 | 16 | 64 | Adaptive | $7.269 \pm 0.33$ | 5.0 |
| | | 0.1 | 16 | 64 | Adaptive | $6.710 \pm 0.34$ | 5.0 |
| | SDXL | 0.5 | 16 | 64 | Adaptive | $7.301 \pm 0.24$ | 17.9 |

## C PSEUDOCODES

As an aid, we provide pseudocodes for the design of Demons Algorithm 2, Algorithm 3:

---
**Algorithm 2** Tanh Demon with Adaptive Temperature
---
1: **Input:** A list of ODE reward estimate $[R_k]$
2: **Output:** Noise Weights $[b_k]$
3: $K \leftarrow \textbf{length}([R_k])$
4: $\hat{\mu} \leftarrow \frac{1}{K} \sum_{k=1}^{K} R_k$
5: $\tau \leftarrow \sqrt{\frac{1}{K} \sum_{k=1}^{K} (R_k - \hat{\mu})^2}$
6: **for** $k = 1$ **to** $K$ **do**
7:     $b_k \leftarrow \tanh\left(\frac{R_k - \hat{\mu}}{\tau}\right)$
8: **end for**
9: **Return** $[b_k]$

---

---
**Algorithm 3** Boltzmann Demon with Fixed Temperature $\tau$
---
1: **Input:** A list of ODE reward estimate $[R_k]$
2: **Output:** Noise Weights $[b_k]$
3: $K \leftarrow \textbf{length}([R_k])$
4: $Z \leftarrow \frac{1}{K} \sum_{k=1}^{K} \exp\left(\frac{R_k}{\tau}\right)$
5: **for** $k = 1$ **to** $K$ **do**
6:     $b_k \leftarrow \frac{1}{Z} \exp\left(\frac{R_k}{\tau}\right)$
7: **end for**
8: **Return** $[b_k]$

---

## D MATHEMATICS

### D.1 ERROR COMPREHENSION FOR REWARD ESTIMATE APPROXIMATION

In this section, we present the theoretical analysis and proof to better understand the error in our reward estimate approximation.

#### D.1.1 ERROR TERM AS AN ITÔ INTEGRAL

**Lemma 1.** *Let the reward estimate function, $h(\mathbf{x}_t, t) = (r \circ \mathbf{c})(\mathbf{x}_t, t)$, be shorthanded as h. We have:*

$$r_\beta(\mathbf{x}_t, t) - (r \circ \mathbf{c})(\mathbf{x}_t, t) = \mathbb{E}_{\mathbf{x}_0|\mathbf{x}_t} \left[ \int_t^0 \nabla_{\mathbf{x}_u} h \cdot \mathrm{d}\mathbf{J}_\beta(\mathbf{x}_u, u) - \beta u^2 \nabla^2 h \, \mathrm{d}u \right]. \tag{13}$$

*where $\mathbf{x}_0$ is sampled from Equation (2), $\nabla^2 h$ is the Laplacian of h and*

$$\mathrm{d}\mathbf{J}_\beta(\mathbf{x}_u, u) = -\beta u^2 \nabla_{\mathbf{x}_u} \log p(\mathbf{x}_u, u) \, \mathrm{d}u + \sqrt{2\beta} u \, \mathrm{d}\omega_u, \tag{14}$$

*Proof.* We aim to prove:

$$r(\mathbf{x}_0) - (r \circ \mathbf{c})(\mathbf{x}_t, t) = \int_t^0 \nabla_{\mathbf{x}_u} h \cdot \mathrm{d}\mathbf{J}_\beta(\mathbf{x}_u, u) - \beta u^2 \nabla^2 h \, \mathrm{d}u, \tag{15}$$

Recall that

$$\mathbf{x}_0 = \mathbf{x}_t + \int_t^0 \mathbf{f}_\beta(\mathbf{x}_u, u) \, \mathrm{d}u + g_\beta(u) \, \mathrm{d}\omega_u, \tag{16}$$

$$\mathbf{c}(\mathbf{x}'_t, t) = \mathbf{x}'_t + \int_t^0 \mathbf{f}_0(\mathbf{x}'_u, u) \, \mathrm{d}u. \tag{17}$$

For an ODE trajectory $\mathbf{x}'(t)$, notice that:

$$0 = \frac{\mathrm{d}}{\mathrm{d}t} h(\mathbf{x}'_t, t) = \frac{\partial h}{\partial t} + \nabla_{\mathbf{x}} h \cdot \frac{\mathrm{d}\mathbf{x}'}{\mathrm{d}t} = \frac{\partial h}{\partial t} + \nabla_{\mathbf{x}} h \cdot \mathbf{f}_0. \tag{18}$$

We can write:

$$r(\mathbf{x}_0) - (r \circ \mathbf{c})(\mathbf{x}_t, t) = h(\mathbf{x}_0, 0) - h(\mathbf{x}_t, t) = \int_t^0 \mathrm{d}h, \tag{19}$$

where $\mathbf{x}_t$, which is not an ODE trajectory (noted by $\mathbf{x}'_t$), follows the SDE trajectory. Using Itô's lemma Ito et al. (1951), we find:

$$\mathrm{d}h = \left( \frac{\partial h}{\partial t} + \nabla_{\mathbf{x}} h \cdot \mathbf{f}_\beta - \frac{1}{2} \cdot g_\beta^2 \nabla^2 h \right) \mathrm{d}t + g_\beta \nabla_{\mathbf{x}} h \cdot \mathrm{d}\omega_t \tag{20}$$

$$= \left( \frac{\partial h}{\partial t} + \nabla_{\mathbf{x}} h \cdot \mathbf{f}_\beta - \left( \frac{\partial h}{\partial t} + \nabla_{\mathbf{x}} h \cdot \mathbf{f}_0 \right) - \frac{1}{2} g_\beta^2 \nabla^2 h \right) \mathrm{d}t + g_\beta \nabla_{\mathbf{x}} h \cdot \mathrm{d}\omega_t \tag{21}$$

$$= \left( \nabla_{\mathbf{x}} h \cdot (\mathbf{f}_\beta - \mathbf{f}_0) - \frac{1}{2} g_\beta^2 \nabla^2 h \right) \mathrm{d}t + g_\beta \nabla_{\mathbf{x}} h \cdot \mathrm{d}\omega_t \tag{22}$$

$$= \nabla_{\mathbf{x}} h \cdot \left( -\beta t^2 \nabla_{\mathbf{x}_t} \log p(\mathbf{x}_t, t) \, \mathrm{d}t + \sqrt{2\beta} t \, \mathrm{d}\omega_t \right) - \beta t^2 \nabla^2 h \, \mathrm{d}t. \tag{23}$$

The sign of the Itô correction term is flipped due to reverse time Brownian Motion-—and the other is followed by expansion. We thus derived Equation (15). $\qquad\square$

### D.1.2 DISCUSSION

We interpret the error terms of the reward estimates approximation as follows:

- The estimate becomes more accurate as $\beta$ decreases, satisfying the intuition that SDE trajectories will reduce to the ODE trajectory as $\beta \to 0$.
- If $\nabla_{\mathbf{x}_u} h \perp \nabla_{\mathbf{x}_u} \log p(\mathbf{x}_u, u)$, the term $\nabla_{\mathbf{x}_u} h \cdot \mathrm{d}\mathbf{J}_\beta(\mathbf{x}_u, u)$ cancels out in expectation.
- If $\nabla^2 h \equiv 0$ and the previous condition holds, then $r \circ \mathbf{c} \equiv r_\beta$.

For estimation purposes, we make the following assumptions to facilitate understanding and derivation of Equation (5):

$$\nabla_{\mathbf{x}_t} \log p(\mathbf{x}_t, t) \approx -\frac{\mathbf{x}_t}{t^2} \tag{24}$$

$$\mathbf{c}(\mathbf{x}_t, t) \approx C_t \mathbf{x}_t \tag{25}$$

$$\nabla_{\mathbf{x}} r \perp \mathbf{x} \tag{26}$$

where $C_t$ is a time-dependent constant and $r$ is scale-invariant.

- Equation (24) is derived from the assumption that $p(\mathbf{x}_t) \approx \mathcal{N}(\mathbf{0}, t^2 \mathbf{I})$.
- Equation (25) stems from image preprocessing algorithms, such as those used in Stable Diffusion, which normalize the image distribution. This normalization implies that images in the dataset are often scaled to lie on a sphere. Therefore, we can reasonably assume that a randomly generated $\mathbf{x}_t$ is close to an image in the dataset in direction.
- Equation (26) is based on the intuition that minor changes in brightness do not significantly affect the semantic interpretation of an image. Besides, many training algorithms incorporate scaling as part of data augmentation, which aligns with the assumption that the gradient of $\nabla_{\mathbf{x}} r$ is orthogonal to $\mathbf{x}$.

Under these assumptions, we obtain:

$$\mathrm{d}h = \nabla_{\mathbf{x}} h \cdot \left( -\beta t^2 \nabla_{\mathbf{x}_t} \log p(\mathbf{x}_t, t) \, \mathrm{d}t + \sqrt{2\beta} t \, \mathrm{d}\omega_t \right) - \beta t^2 \nabla^2 h \, \mathrm{d}t \tag{27}$$

$$\approx C_t \nabla_{\mathbf{x}} r \cdot \left( -\beta \mathbf{x}_t \, \mathrm{d}t + \sqrt{2\beta} t \, \mathrm{d}\omega_t \right) - \beta t^2 \nabla^2 h \, \mathrm{d}t \tag{28}$$

$$\approx \sqrt{2\beta} t C_t \nabla_{\mathbf{x}} r \cdot \mathrm{d}\omega_t - \beta t^2 C_t^2 \nabla^2 r \, \mathrm{d}t \tag{29}$$

If $r$ is harmonic, i.e., $\nabla^2 r \equiv 0$, then $\mathrm{d}h$ becomes a martingale (Billingsley, 2017) and:

$$r_\beta(\mathbf{x}_t, t) - (r \circ \mathbf{c})(\mathbf{x}_t, t) \approx \mathbb{E}_{\mathbf{x}_0 | \mathbf{x}_t} \left[ \int_t^0 \sqrt{2\beta} t C_t \nabla_{\mathbf{x}} r \cdot \mathrm{d}\omega_t \right] = 0. \tag{30}$$

The mean value property, an equivalent statement of a harmonic function, states that the value of a harmonic function at any point is the average of its values on any sphere centered at that point. This property provides an intuitive explanation of our method: if $r$ is harmonic, the reward of the ODE-generated image is the mean value of the reward of SDE-generated ones, while empirically, we observe that the ODE generation resembles the SDE variants.

### D.1.3 ILLUSTRATION OF MISMATCH

For better understanding, we provide an example that $r_\beta$ is far from $r \circ \mathbf{c}$. We adopt assumptions in Appendix D.1.2 to illustrate the intuition, and suppose $\mathbf{x}_t$ is a noisy sample at time $t$ such that $\mathbf{c}(\mathbf{x}_t)$ is a sharp local maxima of $r$, where $\nabla^2 r \ll 0$ near $\mathbf{c}(\mathbf{x}_t)$. Further, suppose that $\beta$ is small enough such that the generated $\mathbf{x}_0$ is near $\mathbf{c}(\mathbf{x}_t)$. In this case, $r_\beta(\mathbf{x}_t) - (r \circ \mathbf{c})(\mathbf{x}_t) < 0$ as $r_\beta(\mathbf{x}_t) = \mathbb{E}_{\mathbf{x}_0 | \mathbf{x}_t} [r(\mathbf{x}_0)] < (r \circ \mathbf{c})(\mathbf{x}_t)$ by intuition.

We can also verify $r_\beta(\mathbf{x}_t) - (r \circ \mathbf{c})(\mathbf{x}_t) < 0$ using Equation (15). Under the assumptions in Appendix D.1.2, we can write:

$$r_\beta(\mathbf{x}_t) - (r \circ \mathbf{c})(\mathbf{x}_t) \approx \mathbb{E}_{\mathbf{x}_0|\mathbf{x}_t}\left[\int_t^0 \sqrt{2\beta}tC_t\nabla_{\mathbf{x}}r \cdot \mathrm{d}\omega_t - \beta u^2\nabla^2 h \,\mathrm{d}u\right] \tag{31}$$

$$= \mathbb{E}_{\mathbf{x}_0|\mathbf{x}_t}\left[\int_t^0 -\beta u^2 C_t^2 \nabla^2 r \,\mathrm{d}u\right] \tag{32}$$

$$< 0. \tag{33}$$

Note that the value of $\nabla^2 r$ is taken at $\mathbf{c}(\mathbf{x}_t)$, fluctuating with SDE.

## D.2 MARTINGALE PROPERTY OF REWARD ESTIMATES.

A martingale is a sequence of random variables that maintains a certain property over time Billingsley (2017): the expected future value, given all past values, is equal to the current value; for a fixed SDE, the current reward estimate is the expected value of the reward estimates at the next time step:

**Fact 1.** *For any time step $\Delta < 0$ such that $t > t - \Delta > 0$:*

$$r_\beta(\mathbf{x}_t) = \mathbb{E}_{\mathbf{x}_{t-\Delta}|\mathbf{x}_t}\left[r_\beta(\mathbf{x}_{t-\Delta})\right]. \tag{34}$$

Intuitively speaking, this idea stems from the principles of conditional probability, which tell us that our current prediction of the final score should be the same as the average of all possible future predictions.

*Proof.* This result follows directly from the foundational definition of expectation. For variable $r_\beta(\mathbf{x}_t)$, we have:

$$r_\beta(\mathbf{x}_t) = \mathbb{E}_{\mathbf{x}_0|\mathbf{x}_t}\left[r(\mathbf{x}_0)\right] = \mathbb{E}_{\mathbf{x}_{t-\Delta}|\mathbf{x}_t}\left[\mathbb{E}_{\mathbf{x}_0|\mathbf{x}_{t-\Delta}}\left[r(\mathbf{x}_0)\right]\right] = \mathbb{E}_{\mathbf{x}_{t-\Delta}|\mathbf{x}_t}\left[r_\beta(\mathbf{x}_{t-\Delta})\right]. \tag{35}$$

$\square$

## D.3 TANH DEMON

We provide the theoretical idea behind the development of the algorithm. To start with, there exists a linear relationship between the reward estimate increment from $\mathbf{x}_t$ to $\hat{\mathbf{x}}_{t-\Delta}^{(k)}$ and the injected noise $\mathbf{z}^{(k)}$, which can be derived from Itô's lemma Ito et al. (1951) and Kolmogorov backward equations Kolmogoroff (1931), as follows:

$$r_\beta(\hat{\mathbf{x}}_{t-\Delta}^{(k)}) - r_\beta(\mathbf{x}_t) = g(t)\nabla_{\mathbf{x}_t}r_\beta \cdot \mathbf{z}^{(k)}\sqrt{\Delta} + o(\Delta), \quad \text{where} \quad \hat{\mathbf{x}}_{t-\Delta}^{(k)} = \mathrm{heun}(\mathbf{x}_t, \mathbf{z}^{(k)}, t, \Delta), \tag{36}$$

which can be interpreted from an SDE with the following Lemma.

**Claim 1.** *Let $r_\beta(\mathbf{x}_t, t) = \mathbb{E}_{\mathbf{x}_0|\mathbf{x}_t}[r(\mathbf{x}_0)]$ be the expected future reward at time 0, given the current state $\mathbf{x}_t$ at time $t$. Then, under the SDE:*

$$\mathrm{d}\mathbf{x}_t = \mathbf{f}_\beta \,\mathrm{d}t + g_\beta \,\mathrm{d}\omega_t, \tag{37}$$

*the differential of $r_\beta$ is:*

$$\mathrm{d}r_\beta = g_\beta \,\nabla_{\mathbf{x}_t}r_\beta \cdot \mathrm{d}\omega_t. \tag{38}$$

*Proof.* We begin by introducing a change of variables. Let $s = t_{\max} - t$, so that as $t$ decreases from $t_{\max}$ to 0, $s$ increases from 0 to $t_{\max}$. This allows us to consider a forward-time process with standard Brownian motion $\tilde{\omega}_s$.

Given the original SDE, we can write:

$$\mathrm{d}\mathbf{x}_s = -\mathbf{f}_\beta \,\mathrm{d}s + g_\beta \,\mathrm{d}\tilde{\omega}_s, \tag{39}$$

where $\tilde{\omega}_s$ is the standard Brownian motion.

Now, applying Itô's lemma to $r_\beta(\mathbf{x}_s, s)$:

$$\mathrm{d}r_\beta = \left( \frac{\partial r_\beta}{\partial s} - \mathbf{f}_\beta \cdot \nabla_{\mathbf{x}_s} r_\beta + \frac{1}{2} g_\beta^2 \nabla^2 r_\beta \right) \mathrm{d}s + g_\beta \nabla_{\mathbf{x}_s} r_\beta \cdot \mathrm{d}\tilde{\omega}_s. \tag{40}$$

We aim to prove the Kolmogorov backward equation:

$$\frac{\partial r_\beta}{\partial s} - \mathbf{f}_\beta \cdot \nabla_{\mathbf{x}_s} r_\beta + \frac{1}{2} g_\beta^2 \nabla^2 r_\beta = 0. \tag{41}$$

To do so, we integrate Itô's lemma from $s$ to $t_{\max}$:

$$r_\beta(\mathbf{x}_{t_{\max}}) - r_\beta(\mathbf{x}_s) = \int_s^{t_{\max}} \mathrm{d}r_\beta \tag{42}$$

$$= \int_s^{t_{\max}} \left( \frac{\partial r_\beta}{\partial s'} - \mathbf{f}_\beta \cdot \nabla_{\mathbf{x}_{s'}} r_\beta + \frac{1}{2} g_\beta^2 \nabla^2 r_\beta \right) \mathrm{d}s'$$

$$+ \int_s^{t_{\max}} g_\beta \nabla_{\mathbf{x}_{s'}} r_\beta \cdot \mathrm{d}\tilde{\omega}_{s'}. \tag{43}$$

Since $r_\beta(\mathbf{x}_{t_{\max}})$ is a martingale, by taking the expectation (conditioned on $\mathbf{x}_s$) on both sides, we obtain:

$$0 = \mathbb{E}_{\mathbf{x}_{t_{\max}}|\mathbf{x}_s} \left[ r_\beta(\mathbf{x}_{t_{\max}}) - r_\beta(\mathbf{x}_s) \right] \tag{44}$$

$$= \mathbb{E}_{\mathbf{x}_{t_{\max}}|\mathbf{x}_s} \left[ \int_s^{t_{\max}} \left( \frac{\partial r_\beta}{\partial s'} - \mathbf{f}_\beta \cdot \nabla_{\mathbf{x}_{s'}} r_\beta + \frac{1}{2} g_\beta^2 \nabla^2 r_\beta \right) \mathrm{d}s' \right]$$

$$+ \mathbb{E}_{\mathbf{x}_{t_{\max}}|\mathbf{x}_s} \left[ \int_s^{t_{\max}} g_\beta \nabla_{\mathbf{x}_{s'}} r_\beta \cdot \mathrm{d}\tilde{\omega}_{s'} \right]. \tag{45}$$

The expectation of the stochastic integral is zero, as Itô integrals have a mean of zero:

$$\mathbb{E}_{\mathbf{x}_{t_{\max}}|\mathbf{x}_s} \left[ \int_s^{t_{\max}} g_\beta \nabla_{\mathbf{x}_{s'}} r_\beta \cdot \mathrm{d}\tilde{\omega}_{s'} \right] = 0. \tag{46}$$

Thus, we are left with:

$$\mathbb{E}_{\mathbf{x}_{t_{\max}}|\mathbf{x}_s} \left[ \int_s^{t_{\max}} \left( \frac{\partial r_\beta}{\partial s'} - \mathbf{f}_\beta \cdot \nabla_{\mathbf{x}_{s'}} r_\beta + \frac{1}{2} g_\beta^2 \nabla^2 r_\beta \right) \mathrm{d}s' \right] = 0. \tag{47}$$

Since the expectation is zero for any interval $[s, t_{\max}]$, the integrand itself must be zero:

$$\frac{\partial r_\beta}{\partial s} - \mathbf{f}_\beta \cdot \nabla_{\mathbf{x}_s} r_\beta + \frac{1}{2} g_\beta^2 \nabla^2 r_\beta = 0. \tag{48}$$

Thus, the differential of $r_\beta$ is given by:

$$\mathrm{d}r_\beta = g_\beta \nabla_{\mathbf{x}_s} r_\beta \cdot \mathrm{d}\tilde{\omega}_s, \tag{49}$$

Returning to the original time variable $t$, we substitute $s = t_{\max} - t$ yielding:

$$\mathrm{d}r_\beta = g_\beta \nabla_{\mathbf{x}_t} r_\beta \cdot \mathrm{d}\omega_t, \tag{50}$$

completing the proof.

$\square$

Although $g_\beta \nabla_{\mathbf{x}_t} r_\beta$ is inaccessible without distillation and thus an intractable static vector, we can still leverage the linear relationship to derive applications. Using our standard approach of interpreting $r \circ \mathbf{c}$ as $r_\beta$ and recognizing that $r_\beta(\mathbf{x}_{t-\Delta})$ is an unbiased estimator of $r_\beta(\mathbf{x}_t)$ (from Appendix D.2), we practically interpret Equation (36) as:

$$(r \circ \mathbf{c})(\hat{\mathbf{x}}_{t-\Delta}^{(k)}) - \hat{\mu} \approx g_\beta(t) \nabla_{\mathbf{x}_t} r_\beta \cdot \mathbf{z}^{(k)} \sqrt{\Delta}, \quad \text{where} \quad \hat{\mu} = \frac{1}{K} \sum_{k=1}^{K} (r \circ \mathbf{c})(\hat{\mathbf{x}}_{t-\Delta}^{(k)}). \quad (51)$$

From Equation (36), flipping the sign of $\mathbf{z}^{(k)}$ reverses its contribution to $r_\beta$. Therefore, based on the observation $(r \circ \mathbf{c})(\hat{\mathbf{x}}_{t-\Delta}^{(k)}) - \hat{\mu}$, we flip $\mathbf{z}^{(k)}$ accordingly. We show the theoretical analysis and proof for the error of the reward estimate of our Tanh Demon in the following.

**Lemma 2.** *Assume the truncation error terms in Equation* (36) *are negligible and $r_\beta \equiv r \circ \mathbf{c}$. Let $\mathbf{z}^*$ be derived from Equation* (11). *With probability* 1, *$r(\hat{\mathbf{x}}_0^{\mathrm{tanh}}) > r_\beta(\mathbf{x}_t)$, where $\hat{\mathbf{x}}_0^{\mathrm{tanh}}$ is derived by applying $\mathbf{z}^*$ on every step.*

Let $\ell = g_\beta \nabla_{\mathbf{x}_t} r_\beta$. Recall that we assume

$$r_\beta(\hat{\mathbf{x}}_{t-\Delta}^{(k)}) - r_\beta(\mathbf{x}_t) = \ell \cdot \mathbf{z}^{(k)} \sqrt{\Delta} \tag{52}$$

$$r_\beta(\hat{\mathbf{x}}_{t-\Delta}^{\mathrm{tanh}}) - r_\beta(\mathbf{x}_t) = \ell \cdot \mathbf{z}^* \sqrt{\Delta} \tag{53}$$

$$\hat{\mathbf{x}}_{t-\Delta}^{(k)} = \mathrm{heun}(\mathbf{x}_t, \mathbf{z}^{(k)}, t, \Delta) \tag{54}$$

$$\hat{\mathbf{x}}_{t-\Delta}^{\mathrm{tanh}} = \mathrm{heun}(\mathbf{x}_t, \mathbf{z}^*, t, \Delta) \tag{55}$$

$$\mathbf{z}^* = \sqrt{N} \, \mathrm{normalized} \left( \sum_{i=1}^{K} b_k^{\mathrm{tanh}} \mathbf{z}^{(k)} \right) \tag{56}$$

$$b_k^{\mathrm{tanh}} = \tanh \left( \frac{r_\beta(\hat{\mathbf{x}}_{t-\Delta}^{(k)}) - r_\beta(\mathbf{x}_t)}{\tau} \right). \tag{57}$$

We aim to prove the sufficient condition: $r_\beta(\hat{\mathbf{x}}_{t-\Delta}^{\mathrm{tanh}}) > r_\beta(\mathbf{x}_t)$ for each numerical step. Under a rotation of basis, without loss of generality, we assume $\ell$ only has value in the first component, i.e., $\ell = (\ell_1, 0, \ldots, 0)$ and $\ell_1 > 0$. We have:

$$r_\beta(\hat{\mathbf{x}}_{t-\Delta}^{\mathrm{tanh}}) > r_\beta(\mathbf{x}_t) \iff \ell_1 z_1^* \sqrt{\Delta} > 0 \tag{58}$$

**Claim 2.** *With probability* 1, *the first component $z_1^*$ of $\mathbf{z}^*$ is positive.*

*Proof.* Since

$$b_k^{\mathrm{tanh}} = \tanh \left( \frac{r_\beta(\hat{\mathbf{x}}_{t-\Delta}^{(k)}) - r_\beta(\mathbf{x}_t)}{\tau} \right) \tag{59}$$

$$= \tanh \left( \frac{\ell \cdot \mathbf{z}^{(k)} \sqrt{\Delta}}{\tau} \right) \tag{60}$$

$$= \tanh \left( \frac{\ell_1 z_1^{(k)} \sqrt{\Delta}}{\tau} \right), \tag{61}$$

where $z_1^{(k)}$ is the first component of $\mathbf{z}^{(k)}$.

Almost surely, $z_1^{(k)} \neq 0$, so $b_k^{\mathrm{tanh}}$ will have the same sign as $z_1^{(k)}$. This implies $b_k^{\mathrm{tanh}} z_1^{(k)} > 0$.

Since the first component of $\mathbf{z}^*$ will have the same sign as the first component of $\sum_{k=1}^{K} b_k^{\mathrm{tanh}} \mathbf{z}^{(k)}$ i.e. $\sum_{k=1}^{K} b_k^{\mathrm{tanh}} z_1^{(k)} > 0$. We conclude that $z_1^* > 0$.

$\square$

In addition, we provide proof of the linear relationship presented in Equation (36).

## D.4 BOLTZMANN DEMON

Recall that

$$\mathbf{x}_{t-\Delta} := \mathbf{x}_t + \int_t^{t-\Delta} \mathbf{f}_\beta(\mathbf{x}_u, u)\, \mathrm{d}u + g_\beta(u)\, \mathrm{d}\omega_u \tag{62}$$

$$\tilde{\mathbf{x}}_{t-\Delta} := \mathbf{x}_t - \mathbf{f}_\beta(\mathbf{x}_t, t)\Delta + g_\beta(t)\mathbf{z}\sqrt{\Delta} \tag{63}$$

$$\hat{\mathbf{x}}_{t-\Delta} := \mathbf{x}_t - \frac{1}{2}\left[\mathbf{f}_\beta(\mathbf{x}_t, t) + \mathbf{f}_\beta(\tilde{\mathbf{x}}_{t-\Delta}, t-\Delta)\right]\Delta + \frac{1}{2}\left[g_\beta(t) + g_\beta(t-\Delta)\right]\mathbf{z}\sqrt{\Delta} \tag{64}$$

We first present the theoretical analysis and proof for the reward estimate error of the proposed Boltzmann Demon as follows.

**Lemma 3.** *Assume $t$ is bounded by $t_{\max}$ and $r_\beta$ is $L$-Lipschitz. Given $\mathbf{x}_t$, if the truncation error per Heun's SDE step in Equation (64) is $\mathbf{x}_{t-\Delta} = \hat{\mathbf{x}}_{t-\Delta} + o(\Delta)$ as $\Delta \to 0^+$, then we have:*

$$\mathbb{E}\left[r(\hat{\boldsymbol{x}}_0^{\mathrm{boltz}})\right] \geq r_\beta(\boldsymbol{x}_t) - o(L \cdot t_{\max}), \tag{65}$$

*where the expectation denotes that each step of the numerical approximation from every $t$ to $t + \Delta$ is taken with the maximum value of $r_\beta(\cdot)$ among i.i.d. SDE samples $\hat{\boldsymbol{x}}_{t+\Delta}^{(k)}$, representing the Boltzmann Demon with $\tau = 0$.*

Lemma 3 establishes a lower bound based on the sample maximum and reward estimate accuracy, providing an improvement guarantee of expected reward in expectation.

We first claim the following statement.

**Claim 3.**
$$\mathbb{E}\left[r_\beta(\hat{\mathbf{x}}_{t-\Delta}^{\mathrm{boltz}})\right] \geq r_\beta(\mathbf{x}_t) - o(L \cdot \Delta). \tag{66}$$

The rest is the induction of SDE time steps $t_0 = t > \cdots > t_{T-2} > t_{T-1} > t_T = 0$, i.e.,

$$\mathbb{E}\left[r(\hat{\mathbf{x}}_0^{\mathrm{boltz}})\right] = \mathbb{E}\left[r_\beta(\hat{\mathbf{x}}_0^{\mathrm{boltz}})\right] \tag{67}$$

$$\geq \mathbb{E}\left[r_\beta(\hat{\mathbf{x}}_{t_{T-1}}^{\mathrm{boltz}})\right] - o(L \cdot t_{T-1}) \tag{68}$$

$$\geq \mathbb{E}\left[r_\beta(\hat{\mathbf{x}}_{t_{T-2}}^{\mathrm{boltz}})\right] - o(L \cdot (t_{T-1} + (t_{T-2} - t_{T-1}))) \tag{69}$$

$$\vdots \tag{70}$$

$$\geq \mathbb{E}\left[r_\beta(\hat{\mathbf{x}}_t^{\mathrm{boltz}})\right] - o(L \cdot t) \tag{71}$$

$$\geq r_\beta(\hat{\mathbf{x}}_t^{\mathrm{boltz}}) - o(L \cdot t_{\max}) \tag{72}$$

*Proof.* We list the premise as the following:

$$\hat{\mathbf{x}}_{t-\Delta}^{(k)} = \mathrm{heun}(\mathbf{x}_t, \mathbf{z}^{(k)}, t, \Delta) \tag{73}$$

$$\hat{\mathbf{x}}_{t-\Delta}^{(k)} = \mathbf{x}_{t-\Delta}^{(k)} - o(\Delta) \tag{74}$$

$$r_\beta(\mathbf{z}^{\mathrm{boltz}}) = \max\{r_\beta(\hat{\mathbf{x}}_{t-\Delta}^{(1)}), \cdots, r_\beta(\hat{\mathbf{x}}_{t-\Delta}^{(K)})\}. \tag{75}$$

We can deduce that:

$$\mathbb{E}\left[r_\beta(\hat{\mathbf{x}}_{t-\Delta}^{\mathrm{boltz}})\right] = \mathbb{E}\left[\max\{r_\beta(\hat{\mathbf{x}}_{t-\Delta}^{(1)}), \cdots, r_\beta(\hat{\mathbf{x}}_{t-\Delta}^{(K)})\}\right] \tag{76}$$

$$\geq \mathbb{E}\left[r_\beta(\hat{\mathbf{x}}_{t-\Delta}^{(1)})\right] \tag{77}$$

$$= \mathbb{E}\left[r_\beta(\mathbf{x}_{t-\Delta}^{(1)}) - L \cdot o(\Delta)\right] \tag{78}$$

$$= r_\beta(\mathbf{x}_t) - o(L \cdot \Delta) \tag{79}$$

The last equation is followed by Equation (34). Here, $r_\beta(\hat{\mathbf{x}}_{t-\Delta})$ is the numerical estimation of the underlying SDE value $r_\beta(\mathbf{x}_{t-\Delta})$. $\qquad \square$

**Lemma 4.** *When $\tau = \infty$ and the time step is small enough, the Boltzmann Demon sampling is identically distributed as the SDE sampling.*

By adjusting $\tau$, we can smoothly transition from prioritizing high-reward noise samples to the standard SDE sampling method, balancing Demon and SDE strategies; note that when $\tau = \infty$, the weights are $b_k = \exp(0) = 1$. Thus, $\sum_{k=1}^{K} b_k \mathbf{z}_k$ results in a Gaussian distribution $\mathcal{N}(\mathbf{0}, K\mathbf{I}_N)$. This distribution is identical distributed to drawing a Gaussian after both are projected onto a sphere of radius $\sqrt{N}$.

We justify replacing Gaussian sampling with uniform sampling from a sphere of radius $\sqrt{N}$ could result in the same effect of SDE during the Euler-Maruyama discretization of SDEs. Assuming constant drift $\mathbf{f}$ and diffusion $g$ for Euler-Maruyama step, the SDE is $d\mathbf{x} = \mathbf{f}\,dt + g\,d\mathbf{W}$. We aim to demonstrate that this replacement yields an identical distribution under small step sizes. Define:

$$\mathbf{Y}_n = -\mathbf{f}\Delta + \sum_{i=1}^{n} g\sqrt{\frac{\Delta}{n}}\tilde{\mathbf{z}}_i = -\mathbf{f}\Delta + g\sqrt{\Delta}\frac{1}{\sqrt{n}}\sum_{i=1}^{n}\tilde{\mathbf{z}}_i \tag{80}$$

where $\tilde{\mathbf{z}}_i$ are i.i.d. vectors uniformly sampled from the surface of a sphere with radius $\sqrt{N}$ i.e. $\tilde{\mathbf{z}}_i \sim \mathrm{Unif}(\sqrt{N}\mathbb{S}^{N-1})$. Also, define:

$$\mathbf{Y} = -\mathbf{f}\Delta + g\sqrt{\Delta}\mathbf{z} \tag{81}$$

**Claim 4.** $\mathbf{Y}_n$ *converges to* $\mathbf{Y}$ *in distribution as* $n \to \infty$.

*Proof.* To justify replacing Gaussian sampling with uniform sampling from the sphere, it is sufficient to show that the normalized sum converges in distribution to a Gaussian vector $\mathbf{z}$, i.e.

$$\frac{1}{\sqrt{n}}\sum_{i=1}^{n}\tilde{\mathbf{z}}_i \xrightarrow{d} \mathbf{z} \tag{82}$$

Due to the symmetry of the uniform distribution, the expectation of each vector is zero, i.e., $\mathbb{E}[\tilde{\mathbf{z}}_i] = \mathbf{0}$. Moreover, the distribution satisfies $\mathbb{E}\left[\tilde{\mathbf{z}}_i\tilde{\mathbf{z}}_i^\top\right] = \mathbf{I}_N$.

By applying the Central Limit Theorem for vector-valued random variables (see, e.g., Rencher (2005)), we conclude that as $n \to \infty$, the normalized sum converges in distribution to a Gaussian vector $\mathbf{z}$ with mean $\mathbf{0}$ and covariance matrix $\mathbf{I}_N$. It justified, in the limit of $n \to \infty$, the uniform sampling from the sphere replicates the statistical properties of Gaussian sampling in the diffusion term of the original SDE. $\square$

### D.5 HIGH DIMENSIONAL GAUSSIAN ON SPHERE

The original statement is more general in the textbook, but we provide specific proof for Gaussian.

**Lemma 5.** *(Vershynin, 2020, Chap. 3) Let $\mathbf{z}$ be independent and identically distributed (i.i.d.) instances of a standard isotropic Gaussian $\mathcal{N}(\mathbf{0}, \mathbf{I}_N)$ in a high-dimensional space $N$. With a high probability (e.g., 0.9999), it holds that*

$$\|\mathbf{z}\| = \sqrt{N} + \mathcal{O}(1) \tag{83}$$

*Proof.* Consider the norm $\|\mathbf{z}\|^2$, where $\mathbf{z}$ is an instance of a standard isotropic Gaussian $\mathcal{N}(\mathbf{0}, \mathbf{I}_N)$ in $N$ dimensions. The distribution of $\|\mathbf{z}\|^2$ follows a Chi-squared distribution with $N$ degrees of freedom. The mean and variance of this distribution are $N$ and $2N$, respectively.

Applying the central limit theorem argument, we approximate the distribution of $\|\mathbf{z}\|^2$ by a normal distribution when $N$ is large, giving:

$$\|\mathbf{z}\|^2 = N + C\sqrt{N} \tag{84}$$

for some constant $C$, where $C \in \mathcal{O}(1)$ represents fluctuations around the mean which are typically on the order of the standard deviation of $\|\mathbf{z}\|^2$, which is $\sqrt{2N}$.

To connect this with the norm of $\mathbf{z}$, we consider:

$$\lim_{N \to \infty} \sqrt{N + C\sqrt{N}} - \sqrt{N} = \lim_{N \to \infty} \sqrt{N} \left( \sqrt{1 + \frac{C}{\sqrt{N}}} - 1 \right) \tag{85}$$

$$= \lim_{N \to \infty} \sqrt{N} \left( \frac{C}{2\sqrt{N}} \right) \tag{86}$$

$$= \frac{C}{2} \tag{87}$$

Here, we use the Taylor series expansion for $\sqrt{1+x}$, approximated as $1 + \frac{x}{2}$ for small $x$, to find the limit. This expansion leads to the conclusion that $\|\mathbf{z}\| = \sqrt{N} + \mathcal{O}(1)$. □

## E COMPARISON ON PICKSCORE

### E.1 PICKSCORE COMPARISONS.

Since PickScore Kirstain et al. (2023) is trained specifically on generated images, we believe it is a more reliable measure and objective than the aesthetics score. To emphasize the strength of our method, we show how the median PickScore reward function improves across 20 different prompts using our Tanh Demon, as shown in Figure 7a.

Our approach utilizes 1,440 reward queries per sample and achieves a PickScore of 0.253, outperforming other methods alongside reduced computation time (180 minutes for our method vs. 240 minutes for resampling methods due to shortened ODE trajectories). Specifically, we compare our method to:

- **SDXL/SDXL-DPO** Wallace et al. (2023a): A state-of-the-art method for direct preference optimization in diffusion models, which achieves a PickScore of 0.226, while the baseline SDXL reaches 0.222.
- **Diffusion-DPO(1440x)**: A variant that selects the highest quality median PickScore from 1440 samples among 20 prompts, achieving a PickScore of 0.246.
- **SDXL(1440x)**: Similar to the above, but without preference optimization, achieving a PickScore of 0.243.

Additionally, resampling an ODE from $\mathbf{x}_{t_{\max}}$ is crucial in applications where the distribution $\mathbf{x}_{t_{\max}} \mid \mathbf{x}_0$ plays a key role, such as in SDEdit Meng et al. (2022). Resampling methods fail to address such applications, highlighting the advantage of our approach.

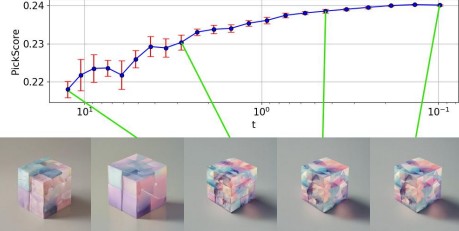
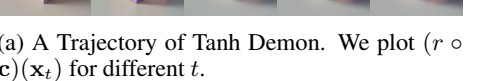
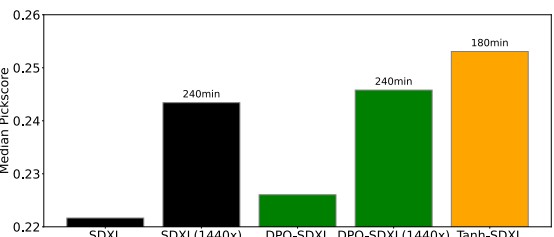

(a) A Trajectory of Tanh Demon. We plot $(r \circ \mathbf{c})(\mathbf{x}_t)$ for different $t$.

(b) The performance of each method on PickScore.

Figure 7: Quantitative results for Tanh Demon.

### E.2 QUALITATIVE RESULTS

In this section, we demonstrate the quantitative and qualitative results of PickScore in SDXL with our Tanh Demon.

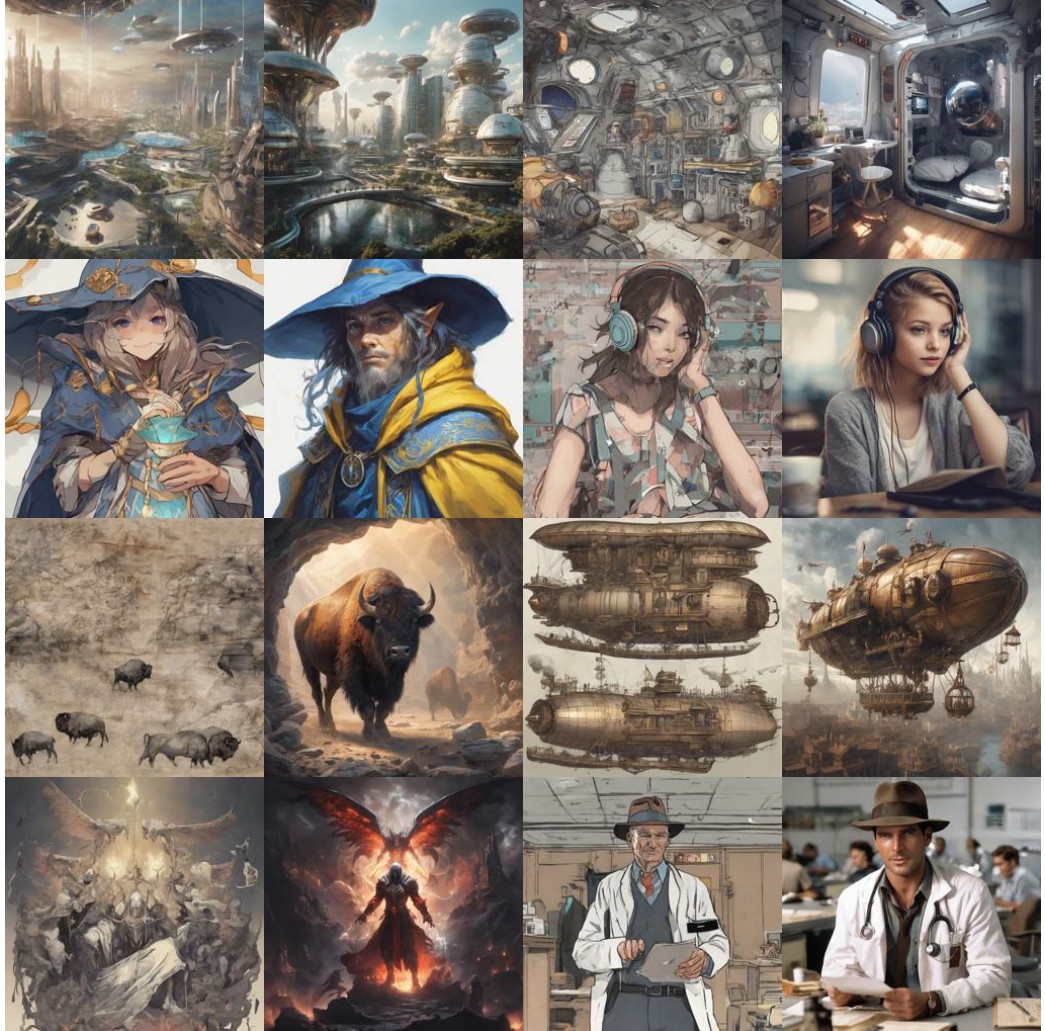

Figure 8: Each row in the figure presents two pairs of images where the image of each pair on the left illustrates results generated using the original PF-ODE method. The image on the right in each pair showcases enhancements achieved by applying our Tanh Demon based on the PickScore metric and SDXL. This figure demonstrates the improvements in visual fidelity and adherence to targeted characteristics achieved through our proposed method.

## F    COMPARISON ON HPSV2

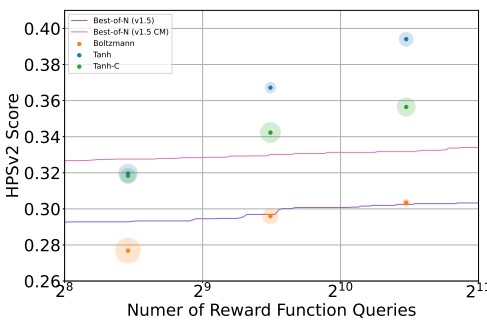
(a) Performance w.r.t Reward Query Number

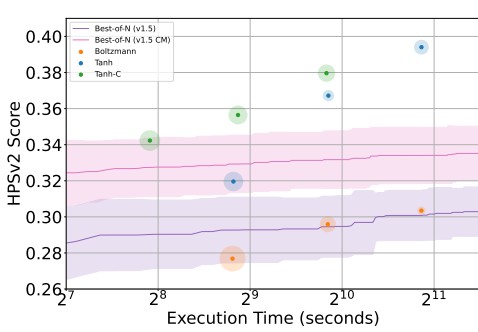
(b) Performance w.r.t Execution Time

Figure 9: Comparison in HPSv2 and HPDv2. The performance comparison of the proposed algorithm and the best-of-N baseline methods is presented in terms of the number of reward queries and execution time, with the dependent variable being $T$. The shaded areas and the solid circle radii represent the evaluation results' standard deviations. If the computational bottleneck is the number of reward queries, we recommend Tanh; if it is computational time, we recommend Tanh-C.

Table 11: We present qualitative results for various methods. For our method, we set $T = 128$, $\beta = 0.5$, and $K = 16$. The Best-of-N samples are generated using 2,336 (5,440 for CM) reward queries and 3.8k seconds, which is significantly more than our method's 1,424 reward queries and 1.8k seconds. Moreover, the presented image from Best-of-N possesses an inferior HPSv2 score compared to ours.

| Best-of-N | Best-of-N (CM) | Tanh-C | Tanh | Boltzmann |
|---|---|---|---|---|
| | | *a castle is in the middle of a eurpean city* | | |
| | | *A motorcycle that is sitting in the dirt.* | | |

We present quantitative and qualitative results in Figure 9 and table 11, using 10 prompts sampled from HPDv2 (Wu et al., 2023). Both the diffusion model and CM are implemented and distilled with SD v1.5.

We observe similar results in Figure 4. Regarding reward queries, the Tanh Demon method outperforms Tanh-C, followed by the Boltzman Demon method. Regarding execution time, however, Tanh-C is recommended over the Tanh Demon if computational time is limited.

## G ADDITIONAL RESULTS WITH VARIOUS REWARD FUNCTIONS.

### G.1 IMAGE GENERATION RESULTS WITH DIFFERENT REWARD FUNCTIONS

We show more image generation results in SDv1.4/SDXL with our Tanh Demon and other reward functions in Tables 12 to 15, using the four reward as objective.

### G.2 SUBJECTIVE TEST OVERVIEW

We surveyed with 101 participants via Google Forms, as shown in Figure 10. Participants evaluated different image generation methods based on:

- **Subjective Preference**: Visual aesthetics and image quality.
- **Semantic Alignment**: Correspondence between generated images and text prompts.

Each participant ranked images across four sections, with rankings aggregated using the following formula:

$$\frac{1}{ML} \sum_{i=1}^{M} \sum_{j=1}^{L} \exp(-(\text{rank}_{ij} - 1)) \tag{88}$$

where:

- $M = 4$ (number of sections),
- $L = 101$ (participants),
- $\text{rank}_{ij}$ is the ranking by participant $j$ for method $i$.

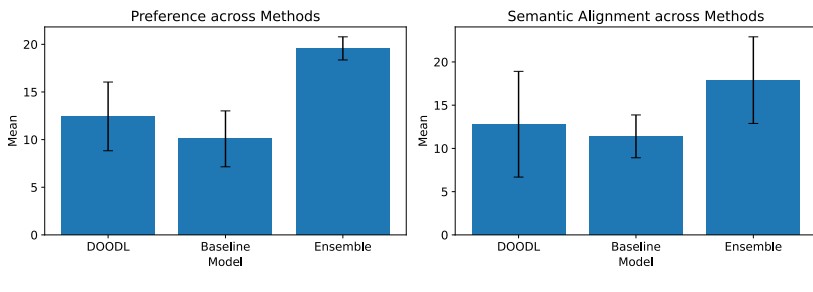

(a) Comparison across methods.

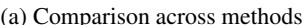

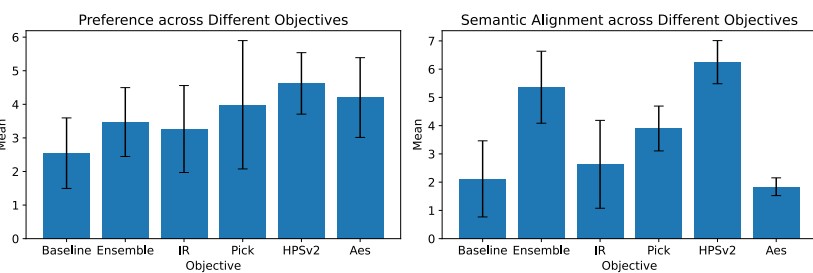

(b) Comparison across objectives.

Figure 10: Subjective test results: Preferences and prompt alignment across methods and objectives.

### G.2.1 SURVEY STRUCTURE

The subjective test comprised four sections: two comparing methods (DOODL, Baseline (SD or SDXL), Ensemble) based on subjective preference and prompt alignment, each with 3 sets

Table 12: Generative Results using SDXL

| SDXL | Aes | IR | Pick | HPSv2 | Ensemble |
|------|-----|-----|------|-------|----------|

An Octopus Playing Chess with a Robot Underwater

A Samurai Gardening on a Floating Island in the Sky

Insanely detailed portrait, wise man

A painting of a girl encountering a giant sunflower blocking her path in a hallway

A demon exiting through a portal

A butterfly flying above an ocean

Table 13: Generative Results using SDXL (Cont)

| SDXL | Aes | IR | Pick | HPSv2 | Ensemble |
|------|-----|-----|------|-------|----------|

Two-faced biomechanical cyborg

Highway to hell

A Jazz Band of Different Alien Species Performing on an Exoplanet

A Victorian Inventor Testing Her Flying Bicycle Above a Steampunk City

A Time Traveler's Picnic at the Edge of a Volcano During the Mesozoic Era

jedi duck holding a lightsaber

Table 14: More Qualitative Results of SD1.4

| Baseline | DOODL | Aes | IR | Pick | HPSv2 | Ensemble |
|---|---|---|---|---|---|---|

An Octopus Playing Chess with a Robot Underwater

Two-faced biomechanical cyborg

Highway to hell

jedi duck holding a lightsaber

A demon exiting through a portal

A Samurai Gardening on a Floating Island in the Sky

Table 15: More Qualitative Results of SD1.4 (cont)

| Baseline | Aes | IR | Pick | HPSv2 | Ensemble | DOODL |
|----------|-----|-----|------|-------|----------|-------|

A Victorian Inventor Testing Her Flying Bicycle Above a Steampunk City

A Time Traveler's Picnic at the Edge of a Volcano During the Mesozoic Era

Insanely detailed portrait, wise man

A butterfly flying above an ocean

A Jazz Band of Different Alien Species Performing on an Exoplanet

A painting of a girl encountering a giant sunflower blocking her path in a hallway

containing one image per method; and two comparing methods applied to different objectives (Baseline, Ensemble, IR, Pick, HPSv2, Aes) also based on preference and prompt alignment, each with 3 sets containing six images per set.

### G.2.2 EVALUATION RESULT OVERVIEW

**Methods Comparison**  Figure 10a shows that DOODL slightly outperforms the Baseline in aesthetic preference and prompt alignment. The Ensemble method significantly surpasses both, indicating superior visual quality and semantic accuracy.

**Objectives Comparison**  As seen in Figure 10b, all objectives outperform the Baseline in prompt alignment, with the HPSv2 method leading. In subjective preference, methods applied to different objectives show varied improvements, with some achieving substantial gains over the Baseline.

### G.2.3 ANALYSIS

We compared DOODL, Baseline, and Ensemble based on aesthetics and prompt alignment. DOODL marginally improves over the Baseline in both criteria, while the Ensemble method consistently outperforms both DOODL and Baseline, excelling in image quality and semantic accuracy. The Ensemble method demonstrates significant enhancements, particularly in tasks requiring visual refinement.

Evaluating different objectives (IR, Pick, HPSv2, Aes) against Baseline and Ensemble revealed that almost all objectives surpass the Baseline in both preference and prompt alignment. However, Aes, an objective without explicit text guidance, shows weaker prompt alignment. Among the objectives, HPSv2 achieves the best performance on both criteria.

The Ensemble method provides the most substantial improvements in visual aesthetics and semantic alignment among method comparisons. Among the factors of the Ensemble method, HPSv2 outperforms other objectives, even the Ensemble method, highlighting its effectiveness in aligning preference for a real human.

## H  MORE DETAILS OF VLM AS DEMON

In this section, we provide more details of experiments and quantitative results of utilizing VLM during generation.

### H.1  GENERATION WITH THE LATEST MODEL

In Table 16, we present an experiment aligning diffusion models to preferences of VLMs from APIs—Google Gemini Flash v1.5 (Gemini Team Google, 2024) and GPT-4o (OpenAI, 2024)—as a demonstration of using non-differentiable reward sources. In each setting, the VLM receives a given scenario, e.g., "You are a journalist who wants to add a visual teaser for your article to grab attention on social media or your news website," and is asked to select the best-matching state vector $\mathbf{x}_{t-\Delta}^{(k)}$ based on generated images. Each scenario is assigned a prompt. In a generation, we apply Tanh Demon with $K = 16$ on SDXL. For scenarios and quantitative results, please refer to Appendix H.

Empirically, VLMs achieve better accuracy when selecting 1 out of 2 options rather than 8 out of 16. We, hence, utilize binary comparisons by applying the *Quicksort partition* (Hoare, 1962) twice on the array of $\mathbf{c}(\mathbf{x}_{t-\Delta}^{(k)})$ derived from $\mathbf{x}_t$. The first application partitions the entire array, and the second partitions the larger subset resulting from the first step. This process allows us to identify roughly the top 8 images within $2K$ comparison.

We assign a $+1$ reward on the roughly top 8 images and $-1$ on the rest for each Demon step. Using PickScore (Kirstain et al., 2023), trained based on image comparisons, we convert logits to probabilities and assess the effectiveness of our method. The results indicate improvements with this metric on every image in any degree.

Table 16: Using VLMs to generate images. PF-ODE (baseline) refers to a baseline without using our method for alignment. The top row of each set indicates the agent's role in the given prompt. The hyperparameters are set to $\beta = 0.1$, $\tau = 0.0001$, $K = 16$, and $T = 128$ using Tanh Demon. We apply ODE after applying 15 Demon steps.

| | Teacher | | | Artist | |
|:---:|:---:|:---:|:---:|:---:|:---:|
| **PF-ODE** | **GPT** | **Gemini** | **PF-ODE** | **GPT** | **Gemini** |
| 13.7% | 17.7% | 68.6% | 24.3% | 27.2% | 48.5% |

| | Researcher | | | Journalist | |
|:---:|:---:|:---:|:---:|:---:|:---:|
| **PF-ODE** | **GPT** | **Gemini** | **PF-ODE** | **GPT** | **Gemini** |
| 9.3% | 69.4% | 21.3% | 32.7% | 33.1% | 34.2% |

| Role | Prompts |
|:---|:---|
| **Teacher** | *A colorful, labeled educational illustration with simple, engaging visuals.* |
| **Artist** | *A detailed, cinematic concept art of unique characters or scenes for games or movies.* |
| **Researcher** | *A realistic, vivid visualization of scientific data with a professional style.* |
| **Journalist** | *A bold, vibrant illustration with dynamic design for digital platforms.* |

## H.2 EXPERIMENTS SETTINGS

We provide the prompt template we used in Table 5 to VLMs. The following are the full system prompts for the scenarios:

Table 17: Roles and System Prompts

| Role | System Prompt |
|:---|:---|
| **Teacher** | *You are a teacher looking to create custom illustrations for your educational materials to make learning more engaging for your students.* |
| **Artist** | *You are a game or movie concept artist tasked with creating concept art for characters, settings, and scenes to speed up the pre-production process.* |
| **Researcher** | *You are a researcher needing to visualize complex data, such as molecular structures in chemistry or weather patterns in meteorology, for better understanding or presentation.* |
| **Journalist** | *You are a journalist who wants to add a visual teaser for your article to grab attention on social media or your news website.* |

Listing 1: Prompt template used in the Gemini image selection task. This template guides the decision-making process for choosing between two images generated from a fixed prompt. For the prompt used in GPT selection, we replace the output format by asking it to return JSON.

```
1  Scenario: {scenario}
2
3  You are presented with two images generated from the prompt "{
     prompt}." Examine both images carefully to decide which one
     best matches the given scenario. Consider how each image
```

```
           relates to the scenario, including its relevance and how well
           it captures the intended theme and concept.
 4
 5 Choose the image that you believe is the most appropriate for the
        scenario. Each image has an id: 0 or 1.
 6
 7 Please share your thought process or any observations you made
        while making your decision. This reflection helps in
        understanding your choice.
 8
 9 Respond with the id of the image you've chosen in a JSON format.
        For example, if you choose the first image, your response
        should look like this:
10
11 ```json
12 [0]
13 ```
14
15 Or, if you choose the second image:
16
17 ```json
18 [1]
19 ```
20
21
22
23 Please response in the following format:
24
25 < Here write down your argument >
26
27 < Here write down your decision, either [0] or [1], the [] is
        necessary >
28
29 Take a deep breath and work on this problem step-by-step. Ensure
        your choice truly aligns with the intended scenario.
```

Listing 2: An example of GPT generated output

```
1 {
2     "justification": "Considering the scenario described, the
        first image is a better choice. This image effectively
        showcases a mysterious, illuminated object that instantly
        attracts the viewer's curiosity. The background hints at
        an academic or historical setting, with architectural
        elements and various artifacts that seem like elements
        from different times and places. This diverse and
        enigmatic setup aligns well with the theme of
        transformation and curiosity across different realms like
        education, history, literature, and science. The glowing
        object in a seemingly ancient, cluttered environment truly
         sparks wonder, making it ideal for grabbing attention on
        social media or a news website.",
3     "chosen_image": [
4          0
5     ]
6 }
```

### H.3 Quantitative Measurement of Effectiveness

For VLMs as reward functions, we use Pickscore Kirstain et al. (2023), which is trained from CLIP Radford et al. (2021), to evaluate the effectiveness of VLM in aligning designed scenarios during image generation. For each scenario, we create a corresponding prompt that partially describes the scenario: "For education" for Teacher, "For entertainment" for Artist, "For research" for Researcher, and "For Journalism" for Journalist. Then, we assess the PickScore between the prompt and the scenario. The results are presented in Table 18, Table 19, Table 20 and Table 21, where the highest score for each prompt is highlighted in bold. Our observations indicate that 14 out of our VLM-generated 16 images demonstrate better PickScore alignment with the corresponding prompt than PF-ODE. Given that all images are generated using the same prompt and initial noisy sample in the same table, these results demonstrate the effectiveness of our approach employing VLM in aligning the scenarios.

Table 18: GPT-SDXL generation, validated by PickScore on related prompts

| Prompt | Teacher | Artist | Researcher | Journalist | PF-ODE |
|---|---|---|---|---|---|
| For education | 0.2050 | 0.2069 | **0.2080** | 0.2071 | 0.2073 |
| For entertainment | 0.2042 | **0.2073** | 0.2061 | 0.2058 | 0.2032 |
| For research | 0.1980 | 0.1989 | **0.1996** | 0.1985 | 0.1971 |
| For journalism | **0.1994** | 0.1957 | 0.1978 | 0.1946 | 0.1970 |

Table 19: Gemini-SDXL generation, validated by PickScore on related prompts

| Prompt | Teacher | Artist | Researcher | Journalist | PF-ODE |
|---|---|---|---|---|---|
| For education | **0.2111** | 0.2042 | 0.2102 | 0.2072 | 0.2073 |
| For entertainment | 0.2057 | 0.2058 | **0.2062** | 0.2013 | 0.2032 |
| For research | 0.2018 | 0.1979 | **0.2035** | 0.1986 | 0.1971 |
| For journalism | 0.2011 | 0.1991 | 0.1978 | **0.2049** | 0.1970 |

Table 20: GPT-SD v1.4 generation, validated by PickScore on related prompts

| Prompt | Teacher | Artist | Researcher | Journalist | PF-ODE |
|---|---|---|---|---|---|
| For education | 0.1978 | **0.2008** | 0.1996 | 0.1988 | 0.1941 |
| For entertainment | **0.2026** | 0.2018 | 0.1991 | 0.2004 | 0.1966 |
| For research | 0.1896 | **0.1936** | 0.1912 | 0.1935 | 0.1878 |
| For journalism | 0.1930 | **0.1951** | 0.1918 | 0.1942 | 0.1901 |

Table 21: Gemini-SD v1.4 generation, validated by PickScore on related prompts

| Prompt | Teacher | Artist | Researcher | Journalist | PF-ODE |
|---|---|---|---|---|---|
| For education | **0.1997** | 0.1945 | 0.1961 | 0.1961 | 0.1941 |
| For entertainment | 0.1954 | 0.1973 | 0.1982 | **0.1989** | 0.1966 |
| For research | **0.1910** | **0.1910** | 0.1897 | 0.1895 | 0.1878 |
| For journalism | 0.1935 | **0.1936** | 0.1927 | 0.1914 | 0.1901 |

## I General Implementation Details

In this section, we show the details of the implementation and experimental settings of the proposed approach as follows.

## I.1 Adapting Stable Diffusion to EDM Framework

In this paper, we tailor the existing text-to-image Stable Diffusion v1.4/v1.5/XL v1.0 (SDXL) (i.e., we use fp16 SD v1.4/SDXL v1.0 for generation.) to the SDE formulation proposed in EDM Karras et al. (2022) by Karras et al. for image generation since its reparameterized time domain, $t \in [t_{\min}, t_{\max}]$, improves numerical stability and sample quality during image generation. We realize the modification through the equation, $\nabla_{\boldsymbol{x}} \log p(\boldsymbol{x}, t) = (D(\boldsymbol{x}, t) - \boldsymbol{x})/t^2$, where the function $D(\boldsymbol{x}, t) = \boldsymbol{x} - t\mathbf{F}(s(t)\boldsymbol{x}, u(t))$ derived from the original model $\mathbf{F}$. In addition, $s(t)$ and $u(t)$ represent the scaling schedule and the original temporal domain of the reparameterized temporal domain $t$, respectively.

## I.2 Numerical Methods for Image Generation

Moreover, for image sampling with ODE/SDE, our approach follows Karras et al. (2022), adopting Heun's method and time intervals determined by $t_i = \left( t_{\max}^{1/\rho} + \frac{i-1}{T-1}(t_{\min}^{1/\rho} - t_{\max}^{1/\rho}) \right)^{\rho}$, setting $\rho = 7, T \geq 20$ and $\ln t_{\max} \approx 2.7, \ln t_{\min} \approx -6.2$. The classifier-free guidance parameter is set to 2 throughout this paper. Across all temporal steps $t$ of image generation, we keep $K$ and $\beta$ constant. We have found that when t is less than 0.11, i.i.d. samples from SDE all appear similar to human perception. For the remaining evaluations, we will directly use ODE. As a result, the actual number of samples will be slightly smaller than $K \cdot T$.

## I.3 Simplifications in Diffusion Process Modification

It is worth noting that in our work, since our main focus is on the modification of the diffusion process, without loss of generality, we omit the VAEs (Kingma & Welling (2014)) of Stable Diffusion models, the prompt $c$, and $\eta$ of classifier-free guidance (CFG) Ho & Salimans (2021) in our formulation for simplicity (i.e., using $p(\boldsymbol{x})$ to denote the unnormalized $p(\boldsymbol{x})p(c \mid \boldsymbol{x})^{\eta}$ for conciseness).

## I.4 Batch Size and Memory Constraints

When we generate many SDE samples, the batch size for solving ODE/SDE is 8 for both Stable Diffusion v1.4, v1.5, and SDXL models. However, due to memory limitations on the RTX 3090, the batch size for evaluating the VAE in SDXL is restricted to 1. This memory bottleneck prevents any further acceleration from using larger batch sizes, as it limits the parallelization during VAE evaluation.

Due to memory limitations, DOODL was run on an Nvidia RTX A6000, which is slightly slower (0.92x) than the RTX 3090 used for the other experiments.

## I.5 Experimental Setup and Hyperparameters

Among all the discussed methods, DOODL is the only one that utilizes the differentiation of the reward function. Due to memory limitations, DOODL was run on an Nvidia RTX A6000, which is slightly slower (0.92x) than the RTX 3090 used for the other experiments. We present the detailed hyperparameter settings of different experiments as follows:

**Baseline Comparison.** The hyperparameters for generation are set to $\beta = 0.5$, $K = 16$, $\eta = 2$ and $\tau$ adaptive for Tanh, $10^{-5}$ for Boltzmann.

**Reward Estimate Approximation Comparison.** We set $\beta = 0.5$ and use SD v1.5 and its distilled CM. The CFG parameter is ignored in CM(set to 1). The reward estimate $r_{\beta}$ is obtained by averaging over 200 Monte Carlo i.i.d. SDE samples—each with 200 SDE steps.

**Generation with Various Reward Functions.** We use Tanh Demon for sampling with adaptive temperature. The hyperparameters for generation are set to $\beta = 0.05$, $K = 16$, $T = 64$ as shown in Tables 4, 12 to 14 and 15 on SD v1.4/SDXL.

For reward scaling in the ensemble setting, the PickScore was multiplied by $98.86$, and HPSv2 was multiplied by $40$.

The interaction step of DOODL is used as suggested by their implementation, $25$ iteration for Aes and $100$ iteration for Pick.

**Non-differentiable Reward.** In Table 5, the hyperparameters are set to $\beta = 0.05$, $\tau = 0.0001$, $K = 16$, and $T = 32$ using Tanh Demon.

**Manual Selection.** In Figure 5, the parameters are $\beta = 0.1, K = 16, T = 128$ but terminate manually, using Tanh Demon with adaptive temperature. We terminate the iteration after ten rounds of operating the UI.

## J    LIMITATIONS

We present the theoretical result in Equation (5), which demonstrates that $r \circ \mathbf{c} \approx r_\beta$. This result relies on the assumption that the reward function $r$ is near harmonic near the ODE sample ourput, as detailed in Appendix D.1.2.

In practice, implementing $r \circ \mathbf{c}$ faces challenges related to time complexity and accuracy bottlenecks, thoroughly discussed in Section 4.

## K    FUTURE WORKS

The only difference between Tanh-C and Tanh Demon lies in how $r \circ \mathbf{c}$ is implemented. Analysis of the data in Table 2 and Figure 4 indicates that Tanh-C's reward performance can be enhanced by mitigating the fidelity in $r \circ \mathbf{c}$ without compromising Tanh-C's speed performance. Potential strategies for improvement include increasing the fidelity of CM distillation or training a dedicated distilled model for $r \circ \mathbf{c}$. We propose these enhancements as future work.

## L    CODE OF ETHICS

The experiments involving human judgment are fully compliant with established ethical standards. Approval is obtained from the Institutional Review Board (IRB) of Academia Sinica under IRB number AS-IRB-HS 02-24031 to ensure that the research meets all necessary guidelines for the ethical treatment of human subjects.

## M    SOCIETAL IMPACT

Our method has the potential to both discourage and encourage harmful content. Users can generate images through manual selections with malicious intentions (Figure 5). This increases accessibility but also raises concerns about misuse. We implement safeguards provided by Stable Diffusion; end-users are responsible for employing them, as recommended in prior works OpenAI (2024); Gemini Team Google (2024); Rombach et al. (2022); Podell et al. (2024), to mitigate potential risks.

