# OpenReview forum: "Training-Free Diffusion Model Alignment with Sampling Demons"
_ICLR.cc/2025/Conference — ICLR 2025 Poster_

### Official Review · Reviewer_tuY1 · 2024-10-29

**Soundness:** 3
**Presentation:** 2
**Contribution:** 3
**Rating:** 6
**Confidence:** 3

**Summary:**

This paper introduces an approach for aligning diffusion models with user preferences at inference time without needing model retraining or backpropagation. The method, named Demon, leverages stochastic optimization to control the noise distribution during the denoising steps of diffusion models, steering the generation process towards user-specified rewards. This paper provides theoretical backing and empirical evidence, including experiments with non-differentiable reward sources such as Visual-Language Model (VLM) APIs and human judgments.

**Strengths:**

1.	This paper presents an interesting approach that eliminates the need for model retraining or backpropagation, which is an advancement in the field of diffusion models and user preference alignment.
2.	The proposed method can be easily integrated with existing diffusion models without further training, making it accessible for immediate application.

**Weaknesses:**

1.	Computational complexity of the proposed method needs to be discussed. While the authors claim ease of integration, the computational complexity of implementing the stochastic optimization approach might be high. I suggest the authors provide the required inference time and GPU memory in Table 3.
2.	Many strong baseline methods have been neglected. I suggest the authors compare with DDPO, DPO, and DPOK in Tables 3 and 4.
3.	The effectiveness of the Demon may be heavily dependent on the quality and fidelity of the reward signal, which may not always be reliable or accurate. Therefore, I suggest the authors conduct experiments to quantitatively compare the results between the human reward signal and a pre-trained reward model.
4.	The writing quality could be further improved. The overall intuition of the proposed method is easy to understand, but the description in Section 4.2 is a bit redundant and complex.

**Questions:**

Please see the Weaknesses.

---

> ### Author Response · Authors · 2024-11-25
> **Thank you for the feedback**
>
> Thank you for your thoughtful review and kind words. We are encouraged that you recognize the importance of eliminating model retraining and backpropagation in our approach. We appreciate your acknowledgment of the **practical accessibility**.
>
> > Computational complexity of the proposed method needs to be discussed. While the authors claim ease of integration, the computational complexity of implementing the stochastic optimization approach might be high. I suggest the authors provide the required inference time and GPU memory in Table 3.
>
> We had addressed the computational complexity in Section 4.2.3 Computational Considerations. Table 3 is **updated with the required inference time.** The GPU requirements of each method are organized in Appendix I. More computational time complexity results is in Table 10.
>
> > Many strong baseline methods have been neglected. I suggest the authors compare with DDPO, DPO, and DPOK in Tables 3 and 4.
>
> In Appendix E, “Comparison on PickScore”, we’ve demonstrated that our method can even **surpass** DPO + Best-of-N method in inference time **under the same inference condition**.
> It could be misleading to add them in Table 3 and Table 4 since DDPO, DPO, and DPOK are **training-based methods**, while ours is an inference-time method. We nevertheless provide a juxtaposition of Diffusion-DPO in **Table 3 and Table 4**.
>
> > The effectiveness of the Demon may be heavily dependent on the quality and fidelity of the reward signal, which may not always be reliable or accurate. Therefore, I suggest the authors conduct experiments to quantitatively compare the results between the human reward signal and a pre-trained reward model.
>
>
> Good point! While we agree that the fidelity of reward signals is very important, we think it’s a universal challenge to all preference alignment methods. Quantitative studies against real humans, which have already been explored in the papers of the reward model themselves e.g., ImageRewad[1] and Pick-a-Pic[2], would be costly and are out of the scope of this paper. We’ll emphasize this research direction in the future as suggested.
>
> [1] Xu, Jiazheng, et al. "Imagereward: Learning and evaluating human preferences for text-to-image generation." Advances in Neural Information Processing Systems 36 (2024).
>
> [2] Kirstain, Yuval, et al. "Pick-a-pic: An open dataset of user preferences for text-to-image generation." Advances in Neural Information Processing Systems 36 (2023): 36652-36663.
>
> > The writing quality could be further improved. The overall intuition of the proposed method is easy to understand, but the description in Section 4.2 is a bit redundant and complex.
>
> We’re encouraged that you find the intuition of the proposed method easy to understand. Thank you for pointing out that Section 4.2 may be redundant and complex. We’ll dive into it to improve the overall reading experience. Any additional suggestions would help us improve its clarity and conciseness.

---

> > ### Comment · Reviewer_tuY1 · 2024-11-26
> >
> > Thanks for your response. Most concerns have been solved except Concern 4. I suggest the authors make Section 4.2 to be more clear and concise.

---

> ### Author Response · Authors · 2024-11-27
>
> Thanks for the response! Given that most concerns are resolved, we are wondering if that would be enough for you to adjust the rating to reflect that?
>
> As for Section 4.2, we would like to have your advice on how to improve the writing. Our initial thought is to focus on simplifying the description around Tanh Demon and potentially adjust/remove figure 3, and add a few overview sentences in the beginning of 4.2 to guide readers. Do you think that would improve the clarity? Or do you have any suggestions?

---

> > ### Comment · Reviewer_tuY1 · 2024-11-27
> >
> > Thanks for your response. I suggest that the authors make Section 4.2 clearer and more concise. Specifically,  the description of Tanh Demon and Boltzmann Demon could be simplified. Introducing Tanh Demon and Boltzmann Demon in paragraphs would be better than subsections. A few overview sentences at the beginning of Section 4.2 would be helpful. I am willing to increase my rating if the above concerns have been solved.

---

> ### Author Response · Authors · 2024-11-27
>
> Thank you for your suggestions. In response, we have made the following modifications to improve the **clarity and conciseness** of Section 4.2.
> First, we added a **brief introductory** statement at the beginning of Section 4.2 to provide an overview and guide readers through the content, as recommended.
> Second, we simplified the **sentence structures** and adopted more accessible wording to enhance readability.
> Third, we **removed some of the mathematical details** from Section 4.2 while retaining them in the appendix for those interested in a deeper understanding.
> Additionally, we moved the subsection on computational considerations (originally 4.2.3) to Section 4.3 to streamline the flow of Section 4.2.
> Finally, we ensured that the introduction of notations follows **a clear, logical order** to avoid confusion. We hope these changes address your concerns and improve the overall reading experience.
>
> Do you think it improves the clarity? Do you have any further suggestions?

---

> > ### Comment · Reviewer_tuY1 · 2024-11-28
> >
> > Thank you for your response. All concerns have been solved. I will increase my rating to 6.

---

> > > ### Author Response · Authors · 2024-11-28
> > >
> > > We are grateful for your feedback and for increasing the rating. Your insights have helped improve our paper.

---

### Official Review · Reviewer_m9Nn · 2024-11-02

**Soundness:** 3
**Presentation:** 3
**Contribution:** 3
**Rating:** 6
**Confidence:** 3

**Summary:**

In this study, the authors develop a method to align the outputs of diffusion models with human preference at inference time. Specifically, the authors address the alignment from the perspective of noise generation. They first propose to measure the quality of noises in each denoising step. Then, they combined different noises to search for an ''optimal'' noise towards a high reward of outputs. Experimental results show that the proposed method effectively improves the aesthetic score of generated images.

**Strengths:**

+ The authors first develop a method to align the generation of diffusion models without training the model or computing the gradient of the reward model.
+ They conducted experiments on various reward models and preference types.
+ The paper is well-organized and easy to follow.

**Weaknesses:**

- Is the proof in this study based on the linear assumption of the reward model? Given that the human preference for images is extremely complicated and certainly nonlinear, does it apply to the scenario of complex human preference labels?
- The noise is flipped in Figure 3, but this operation is neither included in Eq. (11) nor discussed in the main text. Is it a standard operation in your method? On the other hand, I'm not at all convinced that flipping it can ``transform it into high-reward noise’’ because the reward space is not linear and symmetric.
- The comparison between the proposed method and best-of-N is interesting. However, the experiment settings of best-of-N is unclear. I’m not sure whether they selected the final generation with the highest reward or selected the noise at each step. Second, it would be better if the authors could discuss why is the proposed method better than best-of-N based on their theoretical analysis.
- The baseline methods for comparison in experiments are not sufficient. First, they should compare the method with a baseline which always selects the best noise $z^{(k)}$ with the highest reward $(r\odot c)(\cdot)$ at each denoising step. Second, I expect a comparison between the proposed method with more methods based on gradient guidance (Bansal et al. 2024).
- The quantitative results in Section 5 are based on 22 simple prompts of animals, which are not convincing enough.

**Questions:**

Please refer to the above weakness part.

---

> ### Author Response · Authors · 2024-11-25
> **Thank you for the feedback**
>
> Thank you for your kind and thoughtful review. We are encouraged to hear that you found our method **innovative** and appreciated the **organization and clarity** of our paper. Your recognition of our experiments is appreciated.
>
> > Is the proof in this study based on the linear assumption of the reward model?
> > I'm not at all convinced that flipping it can ``transform it into high-reward noise’’ because the reward space is not linear and symmetric.
>
> No, this study is based on the **locally** linear assumption of the reward model. For complicated functions r, we choose a smaller β in Table 9 to confine the neighborhood, thereby ensuring the validity of the linear assumption.
>
> > Given that the human preference for images is extremely complicated and certainly nonlinear, does it apply to the scenario of complex human preference labels?
>
> Yes, our method can be empirically applied to the scenario. In addition, for our **subjective evaluation** of 100 people in Figure 10, the proposed method with different reward functions outperforms DOODL and other baselines in terms of preference and semantic alignment.
>
> > ...I'm not at all convinced that flipping it can ``transform it into high-reward noise’’ because the reward space is not linear and symmetric.
>
> Good question! We want to re-interpreted “transform it into high-reward noise” as it **penalizes low-reward noise**. In Section 4.2.1 of the original paper, "Tanh Demon," flipping the noise z_k is equivalent to multiplying it by a **weight b_k = -1**. We have made it more clear in Ln 285 of the revised paper.
>
> > The comparison between the proposed method and best-of-N is interesting. However, the experiment settings of best-of-N is unclear. I’m not sure whether they selected the final generation with the highest reward or selected the noise at each step.
>
> Best-of-N is the method **for selecting the final generation with the highest reward**: the process iterates many times N. In each iteration, a z_T is drawn from N(0, t^2I). Using the sampled z_T, an image is generated by applying a PF-ODE process. For each point of the x-axis on each plot of Figure 5, we record the mean over the 22 (updated to 45) prompts for the largest reward among the N samples.
>
> > The baseline methods for comparison in experiments are not sufficient. First, they should compare the method with a baseline which always selects the best noise Z with the highest reward (r⊙c)(⋅) at each denoising step.
>
> The method "selecting the noise with highest reward at each step" refers to **Boltzmann Demon with $\tau=0$** in this paper, which is already presented in Figure 5 with an inferior performance than **Tanh Demon**.
>
> > Second, it would be better if the authors could discuss why is the proposed method better than best-of-N based on their theoretical analysis.
>
> By “best-of-N” in this question, if you’re referring to comparing Boltzmann Demon and Tanh Demon, Our explanation is Tanh Demon **penalizes bad noises** (Line 253) and uses them while Botlzmann only uses good noise.
> By “best-of-N” in this question, in case you’re referring to best-of-N in this paper: Demon and Best-of-N are different paradigms empirically; we think it’s not necessary to compare them on a theoretical basis.
>
> > Second, I expect a comparison between the proposed method with more methods based on gradient guidance (Bansal et al. 2024, Universal Guidance of DM).
>
> The proposed method achieves 7.35 in Aes, outperforming Bansal et al. 2024 with 4.11. We also added the results in Table 3, as suggested. However, we would like to highlight the differences in the settings:
>
> To be transparent, we acknowledge some unresolved formulation challenges when diving into Universal Guidance.
>
> A. Universal Guidance and Classifier Guidance (Diffusion Models Beat GANs on Image Synthesis) aim to maximize p(c|x_0) for some categorical classifiers that output logits.
>
> B. Our work focuses on maximizing r(x_0) where r is a reward function that directly evaluates the quality of an image
>
> While A can be implemented within our framework by treating the class logits as the reward r (e.g., Appendix E under orthogonal transformation on the logits), formulating Objective A within the context of B was challenging without introducing unsound assumption e.g. improving the probability of p(Aes=9|x_0). Moreover, we could not identify any implementation details in Universal Guidance that align with B.
> Investigating the compatibility between these objectives is an interesting direction but falls outside the scope of our current study."
>
> Although works in A are not compatible with Figure 4, we still juxtapose **the result of CLIP guidance of Bansal et al. 2024 in Table 4 and Table 3**.

---

> ### Author Response · Authors · 2024-11-25
> **Thank you for the feedback (cont)**
>
> > The quantitative results in Section 5 are based on 22 simple prompts of animals, which are not convincing enough.
>
> Figure 5 is now updated using the **full prompts of DDPO** (Black et al) aligning the prompt settings in DDPO and DRaFT, DPOK. In terms of **reward queries**, our Tanh/Tanh-C outperforms best-of-N and is comparable to backpropagation-based DOODL. In terms of **execution time**, the proposed Tanh/Tanh-C consistently outperforms DOODL due to the exclusion of backpropagation.

---

> > ### Author Response · Authors · 2024-11-27
> >
> > Sorry for following up shortly after our previous response. We are wondering whether it has addressed your concerns sufficiently. If there are additional clarifications you would like to have, we would like to address as soon as we could, as the deadline for updating the manuscript is approaching. Thanks again for the thoughtful feedback.

---

> > > ### Comment · Reviewer_m9Nn · 2024-11-29
> > >
> > > Thank you for your response. I read your response but I find some concerns still remain.
> > >
> > > > "The method "selecting the noise with highest reward at each step" refers to Boltzmann Demon $\tau=0$ with in this paper, which is already presented in Figure 5 with an inferior performance than Tanh Demon."
> > >
> > > However, I could not find this result in Figure 5. I suggest a clearer quantitative comparison of the result like Table 3.
> > >
> > > > "Figure 5 is now updated using the full prompts of DDPO (Black et al) "
> > >
> > > I could not find the corresponding update regarding Figure 5. Moreover, my concern is that all "quantitative results in Section 5", including Table 2 and Table 3, are evaluated on only 22 prompts.
> > >
> > > I understand that authors are not required to add new experiments at this time, but I suggest authors clarify these questions and provide a more solid comparison in the next version of the paper.

---

> ### Author Response · Authors · 2024-11-29
>
> Dear Reviewer m9Nn
>
>
> Thanks for pointing out the issue. From the latest revised pdf, we find that we actually refer to Figure 4 instead of Figure 5. This discrepancy may be caused because some figure and table numbers may have been shifted after pdf update in response to other reviewers’ comments. We apologize for the oversight. We further address other concerns as follows:
>
> 1. **Figure 4**
> * All the lines (*SD v1.5*, *CM v1.5*, *Best-of-N (v1.5)*, *Best-of-N (v1.5 CM)*, *DOODL (v1.5)*, *Boltzmann*, *Tanh*, *Tanh-C*) in **Figure 4** are updated with **the full set**. We **primarily** updated **Figure 4** (*Baseline Comparison*) due to its importance in demonstrating our quantitative effectiveness.
>
>
>
>
>
>
>
>
> 2. **Table 2**
> * We're sorry that we overlooked updating **Table 2** using the full set when we updated the experimental results. **We're currently running experiments to update it with the full set.** We will provide the results here if it is finished before the rebuttal discussion deadline. Otherwise, we will update the results in the final paper. Additionally, we would like to elaborate that this table is used to investigate under what conditions the proximity of $r \circ \mathbf{c} \approx r_\beta$ holds and is less related to the main empirical results.
>
>
>
>
> 3. **Tables 3**
> * To evaluate our method with prompt-aware metrics, e.g., PickScore, the evaluation set is the prompts presented in Tables 12–15, which contain a wide range of prompts (Line 428), instead of animal prompts. Qualitative results are generated using the configuration specified in **Table 3**. We’ll make it more clear in the final paper.
> As for why Boltzmann is not included in **Tables 3**, it is because *Tanh* significantly outperforms *Boltzmann* as shown in Figure 4; however, if you still think it is better to include the results, we’ll rerun the experiments and provide the results here.
>
>
>
>
>
>
>
>
> We hope these address your concerns. Do you have any further questions?

---

> > ### Author Response · Authors · 2024-12-03
> >
> > Dear Reviewer m9Nn,
> >
> >      As the deadline of discussion period (12/3) is approaching, we are wondering whether your concerns are addressed sufficiently. If there are additional clarifications you would like to have, we would like to address as soon as we could. Thank you very much.

---

### Official Review · Reviewer_RS1X · 2024-11-03

**Soundness:** 3
**Presentation:** 3
**Contribution:** 2
**Rating:** 5
**Confidence:** 3

**Summary:**

This paper introduces Demon, an inference-time method for aligning diffusion models with user preferences without retraining or backpropagation. Using stochastic optimization, it guides denoising to focus on high-reward regions, even with non-differentiable rewards like VLM APIs and human judgments. Experiments across three diverse tasks demonstrate improved improvements.

**Strengths:**

1. The concept of analyzing noise quality is interesting
2. The proposed method is a backpropagation-free preference alignment approach that operates during inference, allowing it to be applied flexibly across various conditions.

**Weaknesses:**

1. The comparison includes only a single baseline method, and quantitative results for BoN are missing in Table 3.
2. The evaluation is limited to a small subset of the dataset from DDPO [1]. Expanding the evaluation to include larger datasets, such as the full set from DDPO [1] and the Human Preference Dataset (HPDv2) [2], is recommended.
3. In Table 2, there remains a large performance gap between the 1-step CM and the 6-step ODE. It would be beneficial to consider additional state-of-the-art solvers of diffusion sampling for comparison.

[1] Black, Kevin, et al. "Training diffusion models with reinforcement learning." arXiv preprint arXiv:2305.13301 (2023).

[2] Wu, Xiaoshi, et al. "Human preference score v2: A solid benchmark for evaluating human preferences of text-to-image synthesis." arXiv preprint arXiv:2306.09341 (2023).

**Questions:**

- The authors used the Consistency Model [1] to accelerate computation. Could this model be applied to other diffusion models with different learned data distributions? A more general acceleration method would be beneficial.

[1] Song, Yang, et al. "Consistency models." arXiv preprint arXiv:2303.01469 (2023).

---

> ### Author Response · Authors · 2024-11-25
> **Thank you for the feedback**
>
> Thank you for your kind feedback and thoughtful review. We are encouraged that you found our methodology **interesting**. Your recognition of the **flexibility** of our approach across diverse conditions is deeply appreciated.
>
> > The comparison includes only a single baseline method and quantitative results for BoN are missing in Table 3.
>
> As the first inference-time, backpropagation-free preference alignment method, the only fair baseline is best-of-N. We added the **Best-of-N** method to Table 3. We also provide **DPO** and **Universal Guidance** guided by CLIP for reference.
>
> > The evaluation is limited to a small subset of the dataset from DDPO [1]. Expanding the evaluation to include larger datasets, such as the full set from DDPO [1] and the Human Preference Dataset (HPDv2) [2], is recommended.
>
>
> We updated **Fig. 4 using the full set of DDPO**. In Fig 9 and Table 11, Appendix E, we conducted **another experiment** using sampled prompts from **HPDv2 on HPSv2**.
>
> The results are similar to experiments of aesthetics score: In terms of **reward queries**, our Tanh outperforms best-of-N and other methods. In terms of **execution time**, the proposed Tanh-C achieves an instant improvement over HPSv2 scores.
>
> > In Table 2, there remains a large performance gap between the 1-step CM and the 6-step ODE. It would be beneficial to consider additional state-of-the-art solvers of diffusion sampling for comparison.
>
> As guided by your suggestion, we found that the discrepancy between the versions of our CM (distilled from SDv1.5), which can be regarded as an ODE solver, and the diffusion model (SDv1.4) of our Demon algorithm is the main cause of the gap. By **aligning** their version as SDv1.5, we find Tanh-C can **outperform** Tanh in terms of performance under a fixed period of time. *Many thanks for your suggestion!* We have updated Fig. 4 and Table 2, aligning the diffusion model (v1.5) and CM (distilled from v1.5). We also change $\beta =0.5$ to meet the results of 4.3 Computational Computation.
>
> The justification for adopting Heun’s method is addressed in EDM (Karras et al.) for the trajectory’s low curvature to achieve optimal performance as employing other higher-order solvers.
>
> > The authors used the Consistency Model [1] to accelerate computation. Could this model be applied to other diffusion models with different learned data distributions?
>
> In our case, **Yes**. LCM uses a variety of LAION datasets with a **representative intersection** of Stable Diffusion, slightly biased toward high-aesthetic score data.
>
> > A more general acceleration method would be beneficial.
>
> We found that the underperformance of tanh-C stems from the misalignment of the diffusion model and CM checkpoint. After following your suggestion, tanh-C is sufficiently accelerated (Fig. 4).
>
>
> Karras, Tero, et al. "Analyzing and improving the training dynamics of diffusion models." Proceedings of the IEEE/CVF Conference on Computer Vision and Pattern Recognition. 2024.

---

> > ### Author Response · Authors · 2024-11-27
> >
> > Sorry for following up shortly after our previous response. We are wondering whether it has addressed your concerns sufficiently. If there are additional clarifications you would like to have, we would like to address as soon as we could, as the deadline for updating the manuscript is approaching. Thanks again for the thoughtful feedback.

---

> > > ### Author Response · Authors · 2024-12-03
> > >
> > > Dear Reviewer RS1X,
> > >
> > >       Sorry for following up again as the deadline of discussion period (12/3) is approaching. We are wondering whether our responses have addressed your concerns sufficiently. If there are additional clarifications you would like to have, we would like to address as soon as we could. Thank you very much.

---

### Official Review · Reviewer_fD8w · 2024-11-04

**Soundness:** 2
**Presentation:** 3
**Contribution:** 2
**Rating:** 6
**Confidence:** 3

**Summary:**

The authors present a backpropagation-free, inference-time method for preference alignment in diffusion models. Specifically, the method synthesizes an optimal noise distribution via stochastic optimization thus enhancing the final reward of the generated image.

**Strengths:**

Overall, this paper is thoroughly written, with comprehensive theoretical and empirical analysis. The implementation details are carefully listed, and various qualitative and quantitative metrics are displayed.

**Weaknesses:**

In the “Alignment with Preferences of VLMs” experiment, the role of the VLM doesn’t seem to exhibit a clear preference for a specific style, etc. I understand that the goal is to demonstrate that your method can optimize for a particular non-differentiable preference, such as a VLM API. However, based on the results presented, I’m not convinced that this experiment design effectively shows that your method truly optimizes for a specific preference.

**Questions:**

The paper primarily compares the results with DOODL in terms of computational cost and performance. I would be curious to see how this method compares with other methods mentioned in Table 1 and how it performs with different base models (e.g., SDXL).

Additionally, I would like to know where the performance and computational cost of CM without your method are positioned in Figure 4.

---

> ### Author Response · Authors · 2024-11-25
> **Thank you for the feedback**
>
> Thank you for your kind words and your thoughtful review for highlighting the strengths of our work. We are encouraged to hear that you found our writing **comprehensive**.
>
> > In the “Alignment with Preferences of VLMs” experiment, the role of the VLM doesn’t seem to exhibit a clear preference for a specific style, etc. I understand that the goal is to demonstrate that your method can optimize for a particular non-differentiable preference, such as a VLM API. However, based on the results presented, I’m not convinced that this experiment design effectively shows that your method truly optimizes for a specific preference.
>
> For the original experiments, our paper demonstrates the **efficacy** of a scenario that backpropagation methods, such as VLM through API access, cannot achieve. This claim is supported by a strong **p-value of 0.0021**, which shows that 14 of 16 images have improved on PickScore, a variant of CLIP score. Note that we do not optimize over the CLIP-based measure.
>
>
>
>
>
>
> | Role             | Metric Prompt     | GPT-SDXL ([Role] - [PF-ODE]) / τ | GPT-SDXL PickScore Probability | Gemini-SDXL ([Role] - [PF-ODE]) / τ | Gemini-SDXL PickScore Probability | GPT-SD v1.4 ([Role] - [PF-ODE]) / τ | GPT-SD v1.4 PickScore Prob | Gemini-SD v1.4 ([Role] - [PF-ODE]) / τ | Gemini-SD v1.4 PickScore Probability |
> |:-----------------|:------------------|:-------------------------------------|-------------------:|:---------------------------------------|---------------------:|:-----------------------------------------|---------------------:|:-------------------------------------------|-----------------------:|
> | **Teacher**      | For education     | -0.23   | 0.442752           | 0.38  | **0.593873**             | 0.37   | **0.591459**             | 0.56                                       | **0.636453**               |
> | **Artist**       | For entertainment | 0.41       | **0.601088**           | 0.26     | **0.564636**             | 0.60                                     | **0.645656**             | -0.12    | 0.470036  |
> | **Researcher**   | For research      | 0.09                                 | **0.522485**           | 0.47                                   | **0.615384**             | 0.18                                     | **0.544879**             | 0.32                                       | **0.579324**               |
> | **Journalist**   | For journalism    | 0.24                                 | **0.559714**           | 0.41                                   | **0.601088**             | 0.29                                     | **0.571996**             | 0.34                                       | **0.584191**               |
> | **PickScore (CLIP prob)** | N/A           | N/A                                  | **0.531510**           | N/A                                    | **0.593745**             | N/A                                      | **0.588498**             | N/A                                        | **0.567501**               |
>
>
> We also include more experimental results with **perceptible improvement in Appendix H** by using the more relevant prompt and larger $\beta$ with GPT-4o and Gemini-1.5 as suggested, respectively.
>
>
> > The paper primarily compares the results with DOODL in terms of computational cost and performance.
>
> Good question. The main focus of our work is training-free based methods. DDPO (Black et al.) and other suggested methods are all training-based methods. To the best of our knowledge, DOODL is the closest method to our setting, so we only compare the proposed method with it. However,  we juxtapose the result of diffusion-DPO in **Table 4 as requested**, though they should not be treated as a comparison.
>
> > I would be curious to see how this method compares with other methods mentioned in Table 1 and how it performs with different base models (e.g., SDXL).
>
> We have already provided **qualitative results of SDXL in Table 11** (Table 12 from the latest revision) and Table 12 (from the latest revision) in our initially submitted paper. Also, our experiment Alignment with the preferences of VLMs is actually implemented in SDXL. We’ve made it more clear in Line 444 of the revised paper.
>
> > Additionally, I would like to know where the performance and computational cost of CM without your method are positioned in Figure 4.
>
> We also updated **Figure 4** with an additional Best-of-N of CM.

---

> > ### Author Response · Authors · 2024-11-27
> >
> > Sorry for following up shortly after our previous response. We are wondering whether it has addressed your concerns sufficiently. If there are additional clarifications you would like to have, we would like to address as soon as we could, as the deadline for updating the manuscript is approaching. Thanks again for the thoughtful feedback.

---

### Official Review · Reviewer_UYoG · 2024-11-08

**Soundness:** 3
**Presentation:** 4
**Contribution:** 3
**Rating:** 8
**Confidence:** 3

**Summary:**

This paper studies the problem of aligning diffusion models without additional training or backpropagation. The main idea is to optimize the sampling at inference time, by optimizing the noise used during generation based on a (possibly differentiable) reward signal. To compute the (expected) reward on a given noise, the method leverages a reduction from SDE to ODE dynamics.

**Strengths:**

- very clearly written and polished writing
- clear motivation
- compares to prior methods systematically, and shows consistent improvements

**Weaknesses:**

- It's hard to gauge the significance/improvements in some of the results. For example, Appendix G.2 the numbers are all tightly concentrated around 0.2, so being unfamiliar with the metric, it's hard to know whether the differences are statistically significant or interesting.
- It would be nice to see more experiments with more realistic/extreme use cases of alignment. Most of the variations explored in the paper seem sort of minor. For example, make the model (not) generate NSFW material (or something else distributionally more obvious). Or is there a reason why such examples were note explored?

**Questions:**

- Given that the "noise" is now optimized over, is there a good intuitive way to think about it from a different perspective (diffusion, physics, or bayesian inference, etc.)? (other than the "noise" obviously corresponding to the latent vector)
- My intuition for the proposed sampling strategies improving over "best-of-N" sampling is that now you additionally get to optimize over the simplex spanned by the sampled noise samples (rather than just querying the "vertices" of the simplex).
Could there be more intelligent ways to choose the "basis" vectors for the simplex other than random sample?

---

> ### Author Response · Authors · 2024-11-25
> **Thank you for the feedback**
>
> Thank you for your kind words and thoughtful review for highlighting the strengths of our work. We are pleased to hear that you found our writing **clear and polished**.
>
> > It's hard to gauge the significance/improvements in some of the results. For example, Appendix G.2 the numbers are all tightly concentrated around 0.2, so being unfamiliar with the metric, it's hard to know whether the differences are statistically significant or interesting.
>
> Good advice. We have included the explanation to elaborate the improvement in the paper. For example, as the results in Appendix G.2, we followed the same evaluation setting in diffusion-DPO [1] to use raw PickScore, **a variant of CLIP score**, without scaling as the metric. Although the range of raw scores is close, their relative difference is significant with respect to the temperature (\tau) of 0.01. In addition, when we compute their **p-value** from the fact that 14 of 16 images have improved this metric, the result is 0.0021 which is statistically significant.
>
> Example: The following numbers have the same physical meaning when the temperature is 0.1:
> ```
> logits = [20, 20.1]
> logits = [0, 0.1]
> ```
>
> [1] Wallace, Bram, et al. "Diffusion model alignment using direct preference optimization." Proceedings of the IEEE/CVF Conference on Computer Vision and Pattern Recognition. 2024.
>
> > It would be nice to see more experiments with more realistic/extreme use cases of alignment. ...
>
> Good advice! However, we worry that showing any NSFW material may give rise to an ethical flag.  We will include the results of concept filtering as alternatives in the final paper, as suggested.
>
> > Given that the "noise" is now optimized over, is there a good intuitive way to think about it from a different perspective?
>
> Good question! In fact, our idea to inject biased noise is inspired by **Maxwell’s Demon, a thought experiment in physics,** which is described in Ln 47 of our paper.
>
> > My intuition for the proposed sampling strategies improving over "best-of-N" sampling is that now you additionally get to optimize over the simplex spanned by the sampled noise samples (rather than just querying the "vertices" of the simplex). Could there be more intelligent ways to choose the "basis" vectors for the simplex other than random sample?
>
> Thank you for your insightful explanation. With an **additional projection** to the sqrt N sphere, your description is precise.
> By "basis," we assume you’re referring to orthogonal basis? Few Gaussian samples in a high dimension space are **nearly uncorrelated** with high probability (High-dimensional probability, R Vershynin 2018). Based on the property, the current method design is close to optimal, where each noise z_t sampled suggests a unique direction.

---

> > ### Comment · Reviewer_UYoG · 2024-12-02
> >
> > Thank you for responding to my questions and for clarifications.
> >
> > Re: NSFW:
> > my particular suggestion wasn't about NSFW material per se, but a more distinct shift in distribution with a clear application.
> >
> > Re: interpreting noise and Maxwell's Demon:
> > I understand that there's a loose inspiration, but I was hoping for more of a mechanistic or mathematical interpretation.
> > Perhaps you can view optimizing over the noise as changing the prior of the implicit VAE process defined by diffusion, or something like that.

---

> ### Author Response · Authors · 2024-12-03
>
> Dear Reviewer UYoG,
>
> * We sincerely thank you for your elaborated suggestions. Good advice to show a more distinct shift in distribution! Since we are not able to revise the pdf now,  we will definitely include the application examples, such as the concept filtering, with the proposed method as suggested.
>
> * Additionally, thanks for your feedback. We would like to further elaborate the connection between our proposed method and the Maxwell's Demon. With the proposed reward estimate function, we are able to identify good and bad noises during each sampling step of the diffusion process. Thus, we can encourage the good noises while penalizing the bad ones to synthesize the optimal noise as illustrated in Figure 3. This process is similar to the demon in the Maxwell's Demon thought experiment which controls the gate to let specific molecule pass. We'll make the analogy more clear in the final paper. Do you think that would address your concern?

---

### Author Response · Authors · 2024-11-13
**Acknowledgement**

We sincerely thank all reviewers and the AC for their effort; we're preparing our responses to address each point thoroughly.

---

### Author Response · Authors · 2024-11-25
**Paper Update Summary**

## Dear SAC, AC and Reviewers,

We thank the reviewers for all the insightful comments and address the concerns raised by each reviewer respectively in the following section. The following is a summary of the manuscript we updated in the main paper.

## Update of Figure 4/Table 2

Reviewer RS1X highlighted a performance discrepancy between the ODE solver and the CM solver, resulting in a performance gap between Tanh and Tanh-C. To address this, we aligned the diffusion base model and the CM to version 1.5, as shown in Figure 4 and Table 2, effectively eliminating the performance gap. Additionally, following Reviewer fD8w’s recommendation, we incorporated the best-of-N approach for the CM. In Table 2, we further minimized statistical noise as detailed in Appendix I.5. We set β to 0.5 to ensure consistency with the results presented in Section 4.3, Table 2, and Figure 4. It is important to note that for larger values of β, a higher time step T is necessary to mitigate truncation errors.

## Update of Table 3/Table 4

We add rows of DPO and Universal Guidance and a column of computational requirements (time) in Table 3 and polish the writing according to the added reference.

## Additional Comparison on HPSv2/HPDv2

In appendix F, we add quantitative and qualitative results on HPSv2 to support our method further.

## Additional VLM Result
In Appendix H, Reviewer fD8w found the improvement to be unperceptible. We design a new experiment with larger β and the latest VLM model in Table 16. The new result has a larger margin than Table 5 in terms of PickScore.

---

### Meta-Review · Area_Chair_2tC7 · 2024-12-23

**Metareview:**

This paper proposes a backpropagation-free, inference-time method for preference alignment in diffusion models. Specifically, the authors optimize for the noise distribution to search for an ''optimal'' noise that yields high rewards. Since this method is training-free, it is quite interesting and can be applied to several off the shelf models.

**Additional Comments On Reviewer Discussion:**

During the rebuttal, the authors engaged in several discussions with the reviewers. While the reviewers agree that the novelty of the approach was good, there were several concerns including missing baselines, improving writing quality, etc. The authors addressed most  of the concerns and the paper is in a much better shape now. After the rebuttal, most reviewers lean towards accepting the paper. Hence, I vote for accepting the paper.

---

### Decision · Program_Chairs · 2025-01-22

Accept (Poster)